# Noradrenergic signaling mediates cortical early tagging and storage of remote memory

Xiaocen Fan[1,2,3], Jiachen Song[1,2,3], Chaonan Ma[1,2], Yanbo Lv[1,2], Feifei Wang [1,2], Lan Ma [1,2] ✉ & Xing Liu [1,2] ✉

The neocortical prefrontal memory engram generated during initial learning is critical for remote episodic memory storage, however, the nature of early cortical tagging remains unknown. Here we found that in mice, increased norepinephrine (NE) release from the locus coeruleus (LC) to the medial prefrontal cortex (mPFC) during contextual fear conditioning (CFC) was critical for engram tagging and remote memory storage, which was regulated by the ventrolateral periaqueductal grey. β-Blocker infusion, or knockout of β1-adrenergic receptor (β1-AR) in the mPFC, impaired the storage of remote CFC memory, which could not be rescued by activation of LC-mPFC NE projection. Remote memory retrieval induced the activation of mPFC engram cells that were tagged during CFC. Inhibition of LC-mPFC NE projection or β1-AR knockout impaired mPFC engram tagging. Juvenile mice had fewer LC NE neurons than adults and showed deficiency in mPFC engram tagging and remote memory of CFC. Activation of β1-AR signaling promoted mPFC early tagging and remote memory storage in juvenile mice. Our data demonstrate that activation of LC NEergic signaling during CFC memory encoding mediates engram early tagging in the mPFC and systems consolidation of remote memory.

One of the unique features of memory is that some information can be persistently stored and maintained for weeks or decades long[1]. Remote memory is the basis for our identities and the guide to shape behaviors. Some memories of events that occurred early in life are not able to maintain as those occurring later in life. This infantile amnesia is observed across a wide range of species[2]. The current knowledge of how remote memories are stored, as well as the neural circuits and signal pathways involved, is still scarce.

The standard systems consolidation theory proposes that episodic memory is initially stored within hippocampal circuits and gradually reorganized in the neocortex over time for permanent storage[3,4]. Reversible inactivation of the medial prefrontal cortex (mPFC) impairs retrieval of remote but not recent memory of contextual fear conditioning (CFC)[5], trace fear conditioning[6], trace eyeblink conditioning[7],

and Morris water maze[8]. Remote memory retrieval induces greater c-Fos expression within the mPFC than recent memory retrieval does[9,10]. Remarkably, the revolutionary findings from some recent works reveal the relevance of cortical modules in remote memory storage begins early at the encoding stage. The activation of mPFC engram tagged during or after CFC training is required for the retrieval of remote fear memory, but not for the retrieval of recent fear memory[11,12]. The mPFC engram cells mature, whereas hippocampal engram cells de-mature, after initial memory encoding[11]. The AMPA- and NMDA-receptor-dependent, information-specific early tagging during learning in the orbitofrontal cortex is a crucial neurobiological process for remote memory of social transmission of food preference paradigm[13]. These studies essentially propose the hypothesis that the occurrence of an early activating and strengthening signal in relevant

[1]School of Basic Medical Sciences, State Key Laboratory of Medical Neurobiology, MOE Frontiers Center for Brain Science, Institutes of Brain Science, Department of Neurology, Pharmacology Research Center, Huashan Hospital, Fudan University, Shanghai 200032, China. [2]Research Unit of Addiction Memory, Chinese Academy of Medical Sciences (2021RU009), Shanghai 200032, China. [3]These authors contributed equally: Xiaocen Fan, Jiachen Song. ✉e-mail: lanma@fudan.edu.cn; xingliu@fudan.edu.cn

distributed cortical cell assemblies at the encoding stage supports early cortical engram tagging. This early tagging and gradual maturation of the tagged cortical engram over time might mediate long-lasting memory storage and the process of systems consolidation[14,15]. However, the nature of early cortical tagging remains undefined.

The locus coeruleus (LC) norepinephrine (NE) system plays a broad role in arousal, attention, and cognition[16,17], however, the effects of LC-NE system on remote memory storage are unknown. The mPFC is one of the primary efferent loci of LC. NE release strengthens mPFC functions such as cognitive flexibility and working memory[18,19]. In the present study, we investigated the functional role of long-range LC-mPFC NE projection in remote fear memory storage and memory engram tagging in the mPFC, providing evidence that deficiency in LC-NE system leads to remote memory amnesia.

## Results

### LC-mPFC NE release during memory encoding is required for remote fear memory storage

NE release strengthens mPFC functions such as cognitive flexibility and working memory[18,19]. To understand NE releasing dynamics under fear conditioning, we injected *AAV-hSyn-NE2h* in the mPFC, dentate gyrus (DG), basolateral amygdala (BLA), or nucleus accumbens (NAc), and detected NE release in these brain regions in response to footshock in free-moving mice by recording fluorescence dynamics of NE2h, a NE sensor[20] (Fig. 1a). We observed a concordant increase of NE2h fluorescence in the mPFC, DG, and BLA, but not the NAc, in response to each footshock stimulus (Fig. 1b–m). To determine whether these LC NEergic efferents are required for CFC memory storage, we optogenetically inhibited LC-BLA, LC-DG, or LC-mPFC NE projection of *TH-Cre* mice injected with *AAV-EF1α-DIO-eNpHR3.0-EYFP* or *AAV-EF1α-DIO-EYFP* in the LC, and tested memory on Day 1 (Test 1 for recent memory), Day 14 and 28 (Test 2 and 3 for remote memory) (Fig. 2a). The expression of eNpHR3.0-EYFP or EYFP was detected in LC NE neurons and their terminals in the BLA, DG, and mPFC (Fig. 2b, d, f). Optogenetic inhibition of LC-BLA NE projection during CFC training had no significant effect on the freezing levels in training and memory tests (Fig. 2c), inhibition of LC-DG NE projection during CFC training reduced freezing levels in CFC memory Test 1-3 (Day 1, 14, and 28) (Fig. 2e), and inhibition of LC-mPFC NE projection decreased freezing levels in Test 2 and 3, but not Test 1 (Fig. 2g). Furthermore, to exclude the possible extinction effects produced in Test 1 and Test 2, we conducted a single memory test on Day 28 after training, and found that inhibition of LC-mPFC NE projection impaired memory retention on Day 28 (Fig. 2h, i). Inhibition of the LC-mPFC NE projection on post-conditioning Day 1–7 or 8–14 had no effect on subsequent memory tests and therefore did not affect memory storage (Supplementary Fig. 1). The above results suggest that the activation of LC-mPFC NE projection during CFC encoding is necessarily required for remote memory storage.

Next, we examined the effects of activation of LC-mPFC NE projection on CFC memory storage. We injected *AAV-hSyn-DIO-ChrimsonR-tdTomato* in the LC and *AAV-hSyn-NE2h* in the mPFC of *TH-Cre* mice, and detected NE release in the mPFC in response to each optical activation of LC-mPFC NE projection at 5, 10, 20, and 40 Hz (Fig. 2j, k). The photometry recording showed that 20 Hz laser induced intact fidelity of NE release and its intensity was comparable to those induced by electrical footshock (Fig. 2k, l). 20 Hz laser paired with electrical footshock induced greater NE release in the mPFC than that induced by footshock alone (Fig. 2m). *AAV-EF1α-DIO-hChR2-mCherry* or *AAV-EF1α-DIO-mCherry* was injected in the LC of *TH-Cre* mice. Optogenetic activation of LC-mPFC NE projection (20 Hz) paired with footshock selectively elevated freezing levels in memory Test 2 and 3, but not in Test 1 (Fig. 2n, o). Thus, these data support the notion that the activation of LC-mPFC NE projection during CFC encoding is required for remote memory storage.

### PAG-LC-mPFC projection regulates mPFC NE release and remote fear memory storage

To trace monosynaptic inputs to mPFC, DG, and BLA projecting LC NE neurons, rabies virus-based monosynaptic tracing strategy was employed with the injection of *AAV-EF1α-DIO-TVA-mCherry* mixed with *AAV-EF1α-DIO-RVG* in the LC of *TH-Cre* mice followed by *RV-ENVA-ΔG-eGFP* injection in the mPFC, DG, or BLA two weeks later. Eight days were allowed for rabies to transduce and label synaptically connected input neurons (Fig. 3a). We found that NE[LC-mPFC], NE[LC-DG], and NE[LC-BLA] neurons received large inputs from the ventrolateral periaqueductal gray (vlPAG) (Fig. 3b, c). To determine the role of PAG-LC projection in CFC memory, we optogenetically inhibited vlPAG-LC projection in mice infected with *AAV-hSyn-Cre* combined with *AAV-EF1α-DIO-eNpHR3.0-EYFP* or *AAV-EF1α-DIO-EYFP* in the vlPAG. We found that inhibition of vlPAG-LC projection during training decreased freezing levels in memory tests (Fig. 3d, e), suggesting inhibition of vlPAG-LC projection during encoding impairs both recent and remote CFC memory storage. In addition, inhibition of vlPAG-LC projection significantly decreased NE release in the mPFC in response to footshock (Fig. 3f–h). With an injection of anterograde *scAAV₁-hSyn-FlpO* in the vlPAG and *AAV-hSyn-Con-Fon-eNpHR3.0-EYFP* or *AAV-hSyn-Con-Fon-EYFP* in the LC of *TH-Cre* mice, LC neurons that received input from the vlPAG were allowed expression of eNpHR3.0 (Fig. 3i, j). We found that inhibition of LC-mPFC NE projection innervated by the vlPAG during CFC training decreased freezing levels in Test 2 and 3 (Fig. 3k and Supplementary Fig. 2). According to these results, we propose that vlPAG-LC circuit controlling NE release to different brain regions posits comprehensive roles in memory storage, and LC-mPFC NE release, regulated by vlPAG during CFC encoding, is required for remote fear memory storage.

### LC-mPFC NEergic control of remote memory storage is dependent on β1-AR signaling

To examine the downstream signaling of NE that might mediate remote memory storage, we infused propranolol, a non-selective β-AR antagonist, in the mPFC before CFC (Fig. 4a, b). Propranolol (5 μg/side) treatment did not change the freezing levels in Test 1, but significantly decreased the freezing levels in Test 2 and 3 (Fig. 4c). Nadolol (5 mg/kg, i.p.), a non-selective β-AR antagonist that cannot pass the blood-brain barrier, had no effects on freezing levels in memory tests, precluding the peripheral effects of β-AR on memory storage (Supplementary Fig. 3). We selectively knocked out *Adrb1* or *Adrb2* in the mPFC glutamatergic neurons by injection of *AAV-mCaMKIIα-eGFP-P2A-iCre* in the mPFC of *Adrb1^flox/flox* or *Adrb2^flox/flox* mice, and *Adrb1^+/+* or *Adrb2^+/+* mice were used as the control (Fig. 4d–f, h, i). Deletion of *Adrb1* in the mPFC did not change the freezing levels in Test 1, but decreased freezing levels in Test 2 and 3 (Fig. 4g and Supplementary Fig. 4). *Adrb2* knockout in the mPFC did not change freezing levels in memory tests (Fig. 4j). These data suggest that β1-AR signaling in the mPFC is required for remote memory storage. LC NE projections might co-release NE and DA[21,22], so we deleted *Drd1* in the mPFC glutamatergic neurons by injection of *AAV-mCaMKIIα-eGFP-P2A-iCre* in the mPFC of *Drd1^flox/flox* mice (Fig. 4k, l). Knockout of *Drd1* in the mPFC did not impair memory storage (Fig. 4m), suggesting remote CFC memory storage might be mediated through LC-mPFC NE signaling, but not DA signaling. Furthermore, we injected *AAV-TH-Cre* mixed with *AAV-EF1α-DIO-hChR2-mCherry* in the LC and *AAV-mCaMKIIα-eGFP-P2A-iCre* in the mPFC of *Adrb1^flox/flox* and *Adrb1^+/+* mice (Fig. 4n, o). The results showed that optogenetic activation of LC-mPFC NE projection paired with footshock did not rescue the impairment of remote memory by *Adrb1* knockout in the mPFC (Fig. 4p). Thus, LC-mPFC NE signaling during CFC mediates remote memory storage through β1-AR signaling in the mPFC.

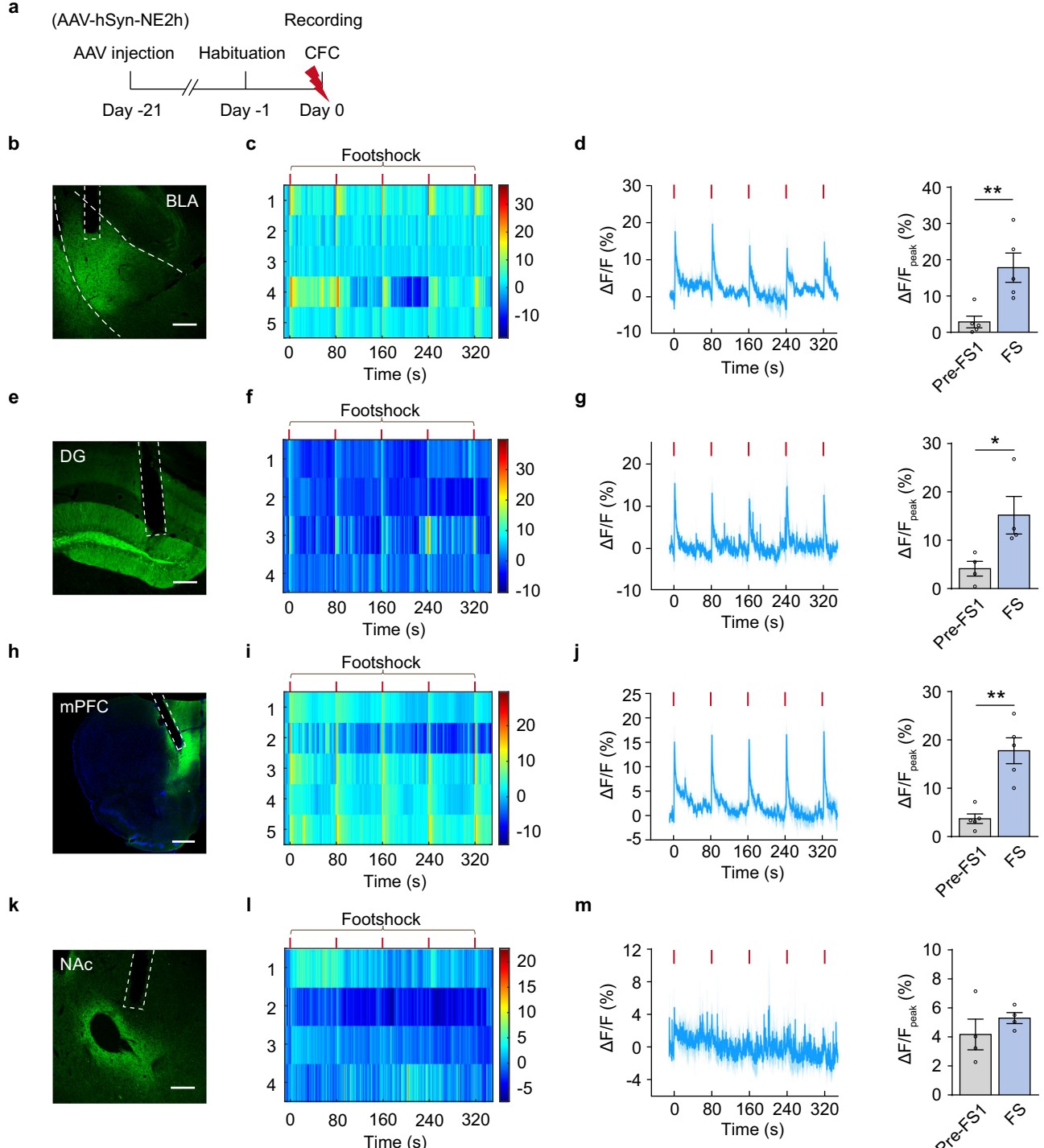

**Fig. 1 | CFC increases NE release in the BLA, DG, and mPFC. a** Experimental scheme. *AAV-hSyn-NE2h* was injected in the BLA, DG, mPFC, or NAc, and NE2h fluorescence was recorded during contextual fear conditioning (CFC). **b**, **e**, **h**, **k** Representative images of NE sensor (NE2h) expression. **c**, **f**, **i**, **l** The heatmaps illustrate the averaged response of NE2h to each footshock (*ΔF/F%*). Each row of heat map represents NE2h fluorescence of a single mouse. Color scale indicates the range of *ΔF/F*. **d**, **g**, **j**, **m** Dynamic response of NE2h to each footshock (Left) and quantitative comparison of mean response to 5 footshocks (Right). The peak values of NE2h fluorescence was calculated within a 5-s window before the first footshock (Pre-FS1) and after each footshock (FS). Scale bar: BLA, DG, and NAc: 200 μm; mPFC: 500 μm. *$p < 0.05$, and **$p < 0.01$ vs indicated group.

## LC-mPFC NE circuit is required for early tagging in the mPFC

The mPFC memory engram cells generated by CFC are critical for remote memory storage[11]. *Fos-TRAP2* or *Arc-TRAP* mice were generated by crossing *Fos$^{2A\text{-}iCreER}$* mice or *Arc$^{CreER}$* mice with a Cre-dependent tdTomato reporter mouse line (AI14). The neurons activated by CFC were labeled with tdTomato under the control of *c-fos* or *Arc* promotor

within a defined time window after tamoxifen induction. Tamoxifen injection did not significantly change locomotor activity in mice when tested 24 h later (Supplementary Fig. 5a, b). Without memory test, c-Fos expression in the mPFC, DG, or BLA was not different 2 days or 14 days after CFC (Fig. 5a–e and Supplementary Fig. 5c–i). Memory tests on Day 14, but not Day 2, induced a significant increase of c-Fos

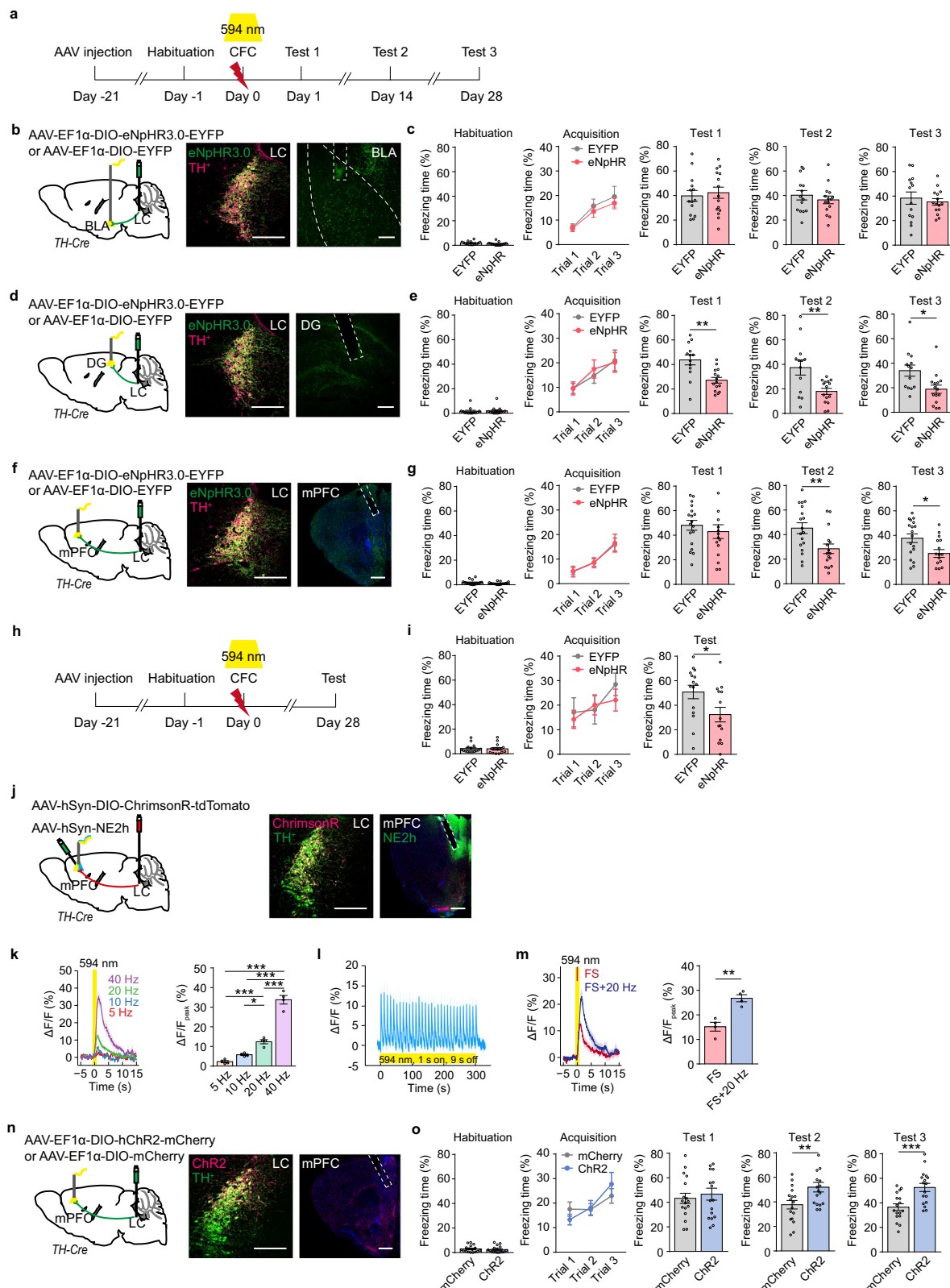

expression in tdTomato$^{+}$ cells in the mPFC (mPFC engram cells), measured as the proportion of tdTomato and c-Fos overlapped cells in the PrL and IL (Fig. 5b–e), indicating remote memory retrieval, but not recent memory retrieval, induces activation of mPFC engram cells. We also found that memory test on Day 2, but not Day 14, increased c-Fos expression in the DG (Supplementary Fig. 5f, g), and memory test on Day 2 and Day 14 induced comparable c-Fos expression in BLA engram cells (Supplementary Fig. 5h, i). Similar results were found in *Arc-TRAP* mice (Supplementary Fig. 6). To determine the role of mPFC engram in memory storage of CFC, we chemogenetically inhibited mPFC engram during memory tests in *c-fos-tTA* mice with injection of *AAV-TRE-tight-hM4Di-mCherry* in the mPFC (Fig. 5f, g). The results showed that inhibition of mPFC engram cells (hM4Di$^{+}$) significantly decreased freezing levels tested on Day 14 and 28 (Fig. 5h), suggesting that activation of

**Fig. 2 | LC-mPFC NE release during CFC is required for remote memory storage.** **a, h** Experimental scheme. *AAV-EF1α-DIO-eNpHR3.0-EYFP* or *AAV-EF1α-DIO-EYFP* was injected in the LC of *TH-Cre* mice, optical fibers were implanted above the BLA, DG, or mPFC. Laser (594 nm) was delivered during CFC, and memory tests were performed 1 day (Test 1), 14 days (Test 2), and 28 days (Test 3) later. **b, d, f** Schematic diagram and representative images of eNpHR3.0-EYFP expression and optical fiber tip. **c, e, g, i** The statistical graphs for freezing levels during habituation, acquisition, and memory tests. **j** Viral injection and representative images of ChrimsonR-tdTomato and NE2h expression and the optical fiber tip in the mPFC. *AAV-hSyn-DIO-ChrimsomR-tdTomato* was injected in the LC, *AAV-hSyn-NE2h* was injected in the mPFC, and an optical fiber was implanted above the mPFC of *TH-Cre* mice. **k** Dynamics of NE sensor in response to 5, 10, 20, 40 Hz laser stimulation (594 nm, 10 mW, 1 s duration) and quantitative comparison of mean responses to each stimulation. **l** Photometry recording showed that 20 Hz laser induced intact fidelity of NE release. **m** Dynamics of NE sensor in response to footshock with or without laser stimulation of ChrimsonR (594 nm, 10 mW, 20 Hz, 1 s duration) and quantitative comparison of mean responses to footshock. Bar graph: the peak values of NE2h fluorescence within a 5-s window after footshock. **n** Viral injection and representative images of ChR2-mCherry expression and the optical fiber tip. *AAV-EF1α-DIO-hChR2-mCherry* or *AAV-EF1α-DIO-mCherry* was bilaterally injected in the LC of *TH-Cre* mice, optical fibers were implanted above the mPFC. **o** The statistical graphs for freezing levels during habituation, acquisition, and memory tests with optogenetic activation (473 nm, 10 mW, 20 Hz, 1 s duration in every 10 s) of LC-mPFC NE projection during footshock. Scale bar: LC, BLA, and DG: 200 μm; mPFC: 500 μm. *$p < 0.05$, **$p < 0.01$ and ***$p < 0.001$ vs indicated group.

mPFC engram cells, generated during memory encoding, is required for remote CFC memory retrieval.

Next, we examined whether LC-mPFC NE projection regulated mPFC engram tagging. The *TH-Cre* mice were injected with *AAV-EF1α-DIO-eNpHR3.0-EYFP* in the LC and *AAV-hSyn-NE2h* in the mPFC. The photometry recording showed that optogenetic inhibition of LC-mPFC NE projection significantly inhibited NE release during CFC (Fig. 5i–l). By injection of *AAV-EF1α-DIO-eNpHR3.0-EYFP* or *AAV-EF1α-DIO-EYFP* in the LC of *TH-Cre* mice, we found that optogenetic inhibition of LC-mPFC NE projection significantly decreased c-Fos expression induced by CFC training (Fig. 5m, n). *Adrb1* knockout in mPFC glutamatergic neurons by injection of *AAV-mCaMKIIα-eGFP-P2A-iCre* in the mPFC of *Adrb1^{flox/flox}* mice also inhibited c-Fos expression induced by CFC training (Fig. 5o, p). These results suggest that NE release and β1-AR mediated signaling are required for mPFC engram tagging during memory encoding.

### Immature LC/NE system and defects of remote memory in juvenile mice

To investigate the roles of LC/NE system in infantile amnesia, we performed CFC task on juvenile mice (20 days old, P20) and young adult mice (70 days old, P70) (Fig. 6a). Juvenile mice showed lower basal locomotor activity and higher basal freezing levels than adult mice in the habituation session (Supplementary Fig. 7). In memory Test 1, the juvenile mice showed similarly high freezing levels as the adult mice. In memory Test 2 to 5, the juvenile mice showed significantly lower freezing levels than those of the adult mice (Fig. 6a), suggesting that the juvenile mice exhibit similar recent memory, but defects in remote memory, compared with adults. The c-Fos immunostaining data showed that CFC increased c-Fos expression in the DG and BLA, but not the mPFC of the juvenile mice (Fig. 6b and Supplementary Fig. 8).

The LC is a tiny nucleus, comprised of only about 1500 neurons in each LC of rats[23]. We estimated the number of LC NEergic neurons in the mice at P14, P21, P28, P42, and P70 by counting the total GFP+ cell numbers in the whole LC of *Dbh::H2B-GFP* and *TH::H2B-GFP* mice. We observed a gradual increase of GFP+ cell counts in the entire LC of the mice from P14 to P70 (Fig. 6c–f and Supplementary Figs. 9, 10). Next, we compared the volume of the LC between juvenile and adult mice with TH immunostaining. We also found a gradual increase in TH volume with similar trends of GFP+ cell counts in the LC during development (Fig. 6g, h, and Supplementary Fig. 11). The LC NE neurons and TH volume remained at lower levels in the mice from P14 to P28 and significantly increased in the adult mice (P70). The fluorescence in situ hybridization (FISH) data revealed a gradual decrease of mRNA expression levels of *Adrb1* and *Adrb2* in the mPFC of the mice from P14 to P70 (Supplementary Fig. 12). These results indicate a deficiency in LC/NE system in juvenile mice. The immature LC/NE system and the failure of mPFC engram tagging might lead to infantile amnesia.

### Adrenergic signaling enhances mPFC early tagging and remote memory storage in juvenile mice

Xamoterol (3 mg/kg, s.c.), a β1-AR agonist, treated 30 min before CFC enhanced c-Fos expression in the mPFC after CFC training and increased freezing levels in Test 2 and 3 of juvenile mice (Fig. 7a–d). Tomoxetine, a norepinephrine reuptake inhibitor, is clinically used for the treatment of attention deficit hyperactivity disorder (ADHD) and depression. Tomoxetine (3 mg/kg, i.p.) treated 30 min before CFC training also elevated c-Fos expression in the mPFC and freezing levels in Test 2 and 3 (Fig. 7e–h). Continuous treatment of xamoterol or tomoxetine for 28 days also significantly increased freezing levels in Test 2 and 3 in juvenile mice (Supplementary Fig. 13). Thus, the activation of NE/β-AR signaling in the mPFC during memory encoding increases engram tagging in the mPFC and promotes remote memory storage in juvenile mice.

## Discussion

In this study, we found that LC-mPFC NE release and β1-AR signaling during CFC were critical for remote memory storage. As illustrated in Fig. 7i, mPFC memory engram generated during initial learning is dependent on NE release and β1-AR signaling in the mPFC. The functional maturation of mPFC engram with time mediates remote memory storage. The juvenile mice have immature LC/NE system and show deficiency in remote memory of CFC. The activation of β1-adrenergic signaling in the mPFC enhances remote memory storage of CFC and memory engram tagging in juvenile mice.

The NE system plays a broad role in fear learning and memory storage. The LC responds to aversive stimuli, and the broad NE release in the forebrain is a broadcast signal. However, how LC NE projections control and organize fear memory storage through its downstream pathways is largely unknown. Our study suggests that LC NE release induced by the electrical footshock regulates fear learning and memory storage at multiple nodes that posit distinct roles. Specifically, LC-mPFC NE release is required for remote memory storage of CFC mediated by β1-AR signaling in the mPFC.

Recent studies point out that the sparsely distributed groups of neurons activated during learning serve as the physical representation of memory trace, suggesting a cellular correlate of memory engram[24]. Our model supports the concept that the mPFC engram is already generated, albeit in an immature form, by initial learning, and subsequently becomes functionally mature with the passage of time. The early tagging hypothesis proposes that the early activation and plasticity occur during learning in the neocortex and the gradual strengthening of neocortical circuits are critical for remote memory recall and systems consolidation[15]. However, the nature of this tagging remains undefined and the underlying mechanisms are largely unknown. Studies show that β-ARs are critically involved in the regulation of synaptic strength. Isoproterenol, the β-AR agonist, potentiates cortical excitatory propagation[25]. Activation of β-ARs enhances memory storage and increases AMPAR- and NMDAR-dependent excitatory synaptic transmission by elevating AC-cAMP-PKA activity[26,27].

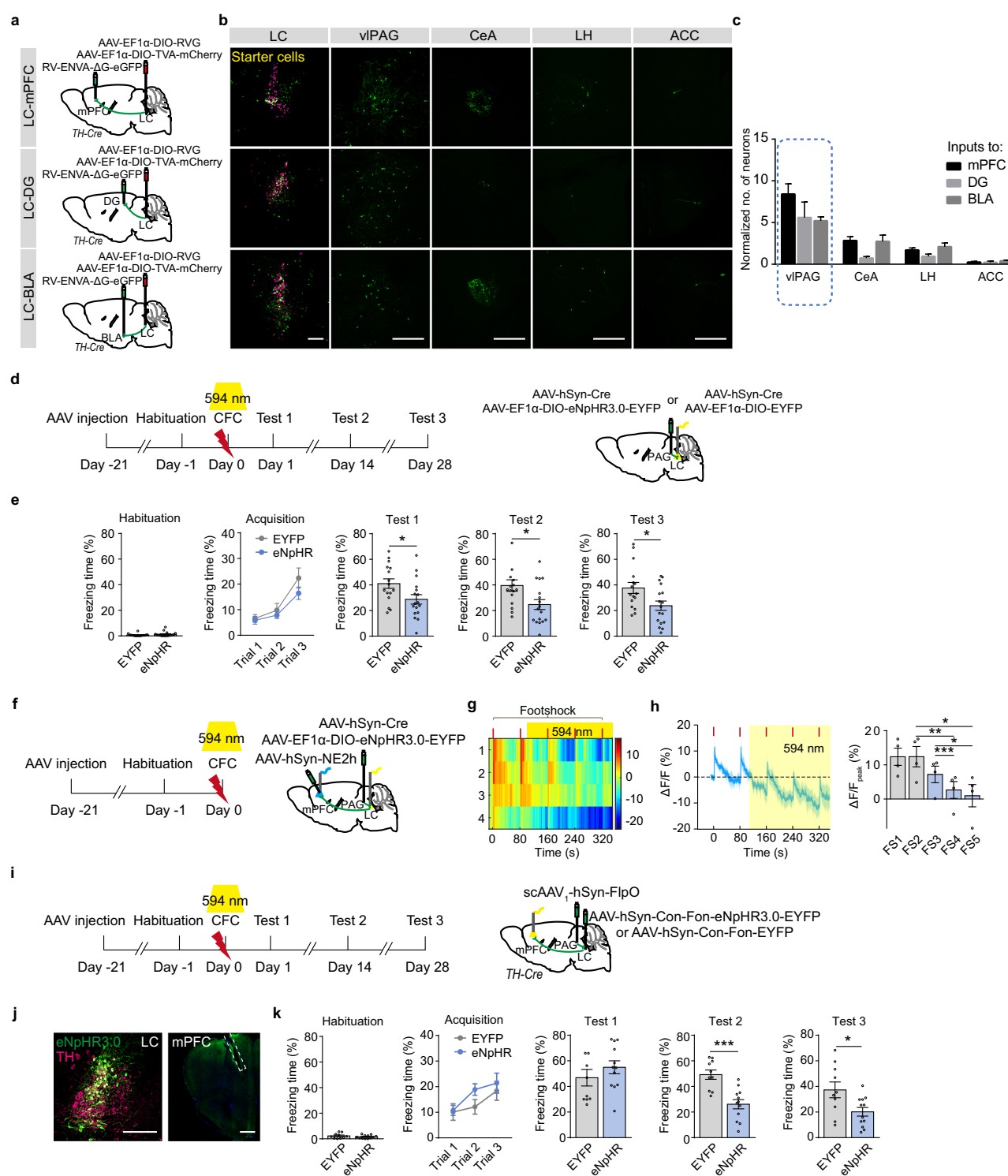

Interestingly, β-AR-dependent regulations of cortical neuronal excitability are reported. β-AR blockade prevents NE-induced potentiation of mPFC neuronal intrinsic excitability[28]. Moreover, the excitability state is an intrinsic physiological property of neurons and also contributes to memory allocation or engram formation[12,29–31]. The precise mechanisms of the increase of intrinsic neuronal excitability during encoding remain unknown. We observed an increase in NE release in response to footshock. β1-AR knockout or antagonism of β-AR in the mPFC during conditioning impaired remote memory, but not recent memory of contextual fear conditioning. Inhibition of NE projection decreased neuronal activation in the mPFC induced by fear

conditioning. Our data indicate that the early engram tagging in the mPFC, critical for remote memory storage, is dependent on NE release and β-AR signaling. The mPFC possesses excitatory projection neurons and a variety of GABAergic interneurons that are engaged by long-range inputs from other brain areas, such as the LC[32]. β1-AR and β2-AR are distributed in mPFC excitatory neurons, but not all types of interneurons[33–35]. Studies show that activation of β-AR or dopamine D1 receptor increases the firing of mPFC excitatory neurons[28,36,37], and facilitates excitatory synaptic transmission in mPFC excitatory neurons[38]. In this study, we found that the expression of *Adrb1*, but not *Adrb2* or *Drd1*, in mPFC excitatory neurons was critical for remote

**Fig. 3 | LC-mPFC NE release innervated by the PAG is required for remote memory storage. a** Experimental scheme. *AAV-EF1α-DIO-TVA-mCherry* mixed with *AAV-EF1α-DIO-RVG* was injected in the LC of *TH-Cre* mice, *RV-ENVA-ΔG-eGFP* was injected in the mPFC, DG, or BLA two weeks later. **b** Representative images of starter cells in the LC and eGFP expression in the vlPAG, CeA, LH, and ACC. **c** Statistical graph for normalized number of neurons that project to NE[LC-mPFC], NE[LC-DG], and NE[LC-BLA] neurons. **d** Experimental scheme. *AAV-hSyn-Cre* mixed with *AAV-EF1α-DIO-eNpHR3.0-EYFP* or *AAV-EF1α-DIO-EYFP* was injected in the vlPAG, and optical fibers were implanted above the LC. Optical stimulation (594 nm, 5 mW) was delivered during CFC and memory tests were performed 1 day, 14 days and 28 days after CFC. **e** Statistical graphs for freezing levels during memory tests. **f** Experimental scheme. *AAV-hSyn-Cre* mixed with *AAV-EF1α-DIO-eNpHR3.0-EYFP* was injected in the vlPAG of wild-type mice, *AAV-hSyn-NE2h* was injected in the mPFC, and optical fibers were implanted above the mPFC and LC. The 594 nm laser

above the LC was delivered during CFC. **g** The heatmap illustrates the averaged response of NE sensor *(ΔF/F%)* to each footshock with or without laser stimulation (594 nm, 5 mW). Each row of heat map represents NE2h fluorescence of a single mouse. Color scale indicates the range of *ΔF/F*. **h** Dynamic response of NE sensor to each footshock and quantitative comparison of the responses to each footshock. Bar graph: The peak values of NE2h fluorescence with a 5-s window after each footshock. **i** Experimental scheme. Anterograde *scAAV₁-hSyn-FlpO* was injected in the vlPAG, *AAV-hSyn-Con-Fon-eNpHR3.0-EYFP* or *AAV-hSyn-Con-Fon-EYFP* was injected in the LC of *TH-Cre* mice, and optical fibers were implanted above the mPFC. The 594 nm laser was delivered during CFC. **j** Representative images of eNpHR3.0-EYFP expression and fiber tip. **k** Statistical graphs for freezing levels during memory tests. Scale bar: 200 μm; mPFC: 500 μm. *$p < 0.05$, **$p < 0.01$, ***$p < 0.001$ vs indicated group.

memory storage. However, whether *Adrb1* expression in other mPFC cell-types, such as GABAergic neurons, regulates memory processing is unclear and deserves further investigation.

In this study, low levels of co-localization between tdTomato[+] and c-Fos[+] cells in the DG were detected after memory retrieval. The DG data are consistent with the previous study that show low activation levels of engram cells (<5%) after memory retrieval in *Arc-TRAP* mice[39]. However, another study shows that activation rate of DG engram cells induced by recent CFC memory retrieval is more than 15% in *c-fos-tTA::TRE-H2B-GFP* mice[11]. This difference may be due to the different efficiency of labeling by TRAP and TetOff systems. In addition, for c-Fos[+] neurons counting, 3–4 coronal slices at 180 μm intervals (every 6 slices) throughout the DG were used, which may not fully represent the engram population in the whole DG.

In general, infantile or childhood amnesia is the inability of adult people to recall episodic memories that occurred before age 3 and the tendency of adults to have sparse recollections before age 10[2,40,41]. The similarities in infantile forgetting have also been reported in animals in remote fear memory[42] and remote spatial memory[43]. One widely accepted hypothesis posits that the hippocampus during infancy is immature and unable to process and store contextual and episodic experiences[44]. The infant mice also show high hippocampal neurogenesis levels and their freshly generated memories tend to be rapidly forgotten[45]. Our data showed that at an early stage, the LC-NE system remained immature. The LC NE cell number of infant mice was significantly lower (P14-21) than that of adult mice (P70). β-AR agonist or NE reuptake inhibitor treated right before CFC recovered "lost" infant memory of contextual fear conditioning. In addition, the mPFC of both humans and rodents develops postnatally and continues to increase in synapse density and maturity[46]. Thus, the immaturation of LC-mPFC NE circuit and lack of NE release at the early stage might account for infantile amnesia. Our results showed that LC NE neurons and TH volume remained at lower levels in the mice from P14 to P28 and significantly increased in the adult mice (P70), suggesting LC NE production and release in juveniles may be less than in adults. Our results showed that mRNA expression levels of *Adrb1* and *Adrb2* were greater in juveniles than adults. However, the difference in mRNA levels of *Adrb1/2* may not translate to the same difference in protein levels and membrane expression. Due to the non-specificity of the commercial primary antibodies to β1-AR and β2-AR we purchased, we did not compare β-AR protein or membrane receptor expression levels in the mPFC between juveniles and adults. We propose that β-AR signaling in the mPFC of juvenile mice is still at low levels, since the c-Fos expression in the mPFC was not significantly increased after fear conditioning. We also found the P20 mice showed significantly lower locomotion and higher freezing levels in the habituation session than P70 mice, suggesting the differences in locomotor activities and anxiety levels between juvenile and adult mice.

The hippocampus is thought to automatically and necessarily encode experiences in infants and adults[45,47]. We found c-Fos

expression was significantly increased in the DG and BLA, but not the mPFC, of juvenile mice after fear conditioning, suggesting at P20, the DG and BLA respond similarly to CFC as the adults, however, the mPFC does not function as the adults. Despite the impaired remote memory of CFC, the juvenile can still form recent CFC memory. It seems memory trace can be established in some brain regions critical for recent memory formation dependent on some neurotransmitter systems or mechanisms besides the immature NE system in juvenile mice. However, immature LC-mPFC NE projection may be one of the reasons that prevent prefrontal engram generation and the transfer of hippocampal memory. Once NE system sufficiently matures, it might positively contribute to memory storage through β-AR signaling in the hippocampus and amygdala, perhaps by the regulation of synaptic plasticity[48–50].

Our data indicate that memory engram tagging in the mPFC during initial learning mediated by LC NE innervation determines the storage of remote CFC memory. These findings point out that LC-mPFC NE circuit bidirectionally regulates remote memory strength and the activation of mPFC β-AR signaling restores infantile amnesia, advancing the understanding of the role of cortical tagging in remote memory storage.

## Methods

### Animals

Adult and juvenile C57BL/6 mice were purchased from Shanghai Laboratory Animal Center, CAS. *TH-Cre* mice (#008601), *R26[AI14]* (AI14) mice (#007914), *Fos[2A-iCreER]* mice (#030323, TRAP2), *Arc[CreER]* mice (#021881), *c-fos-tTA* mice (#018306) and *Drd1[flox/flox]* mice (#025700) were purchased from The Jackson Laboratory (CA, USA). *Dbh-Cre* mice (#036778) were purchased from MMRRC (MO, USA). *Adrb1[flox/flox]* mice and *Adrb2[flox/flox]* mice were developed by our lab. *H2B-GFP* mice were gifts from M He (Institutes of Brain Science, Fudan University, China). *Fos[2A-iCreER]::tdTomato* (*Fos-TRAP2*) and *Arc[CreER]::tdTomato* (*Arc-TRAP*) mice were obtained by cross-breeding *Fos[2A-iCreER/-]* and *Arc[CreER/-]* mice with AI14 mice, separately. *Dbh::H2B-GFP* and *TH::H2B-GFP* mice were generated by crossing *Dbh-Cre* or *TH-Cre* mice with *H2B-GFP* mice. Detailed usages of mouse lines are provided in Supplementary Data 1-Mouse lines and viral vectors.

Mice were housed under standard conditions and maintained in a temperature (22 ± 2 °C) and humidity (50 ± 5%) controlled environment under a 12-h light-dark cycle with ad libitum food and water. Only male mice were used for behavioral experiments. All experiments were conducted in accordance with the National Institutes of Health *Guide for the Care and Use of Laboratory Animals* and all procedures were approved by *the Animal Care and Use Committee of the Shanghai Medical College of Fudan University*.

### Virus Preparation

The AAV-hSyn-NE2h plasmid was a gift from YL Li (Peking University, China), and the virus was packaged by Obio Technology.

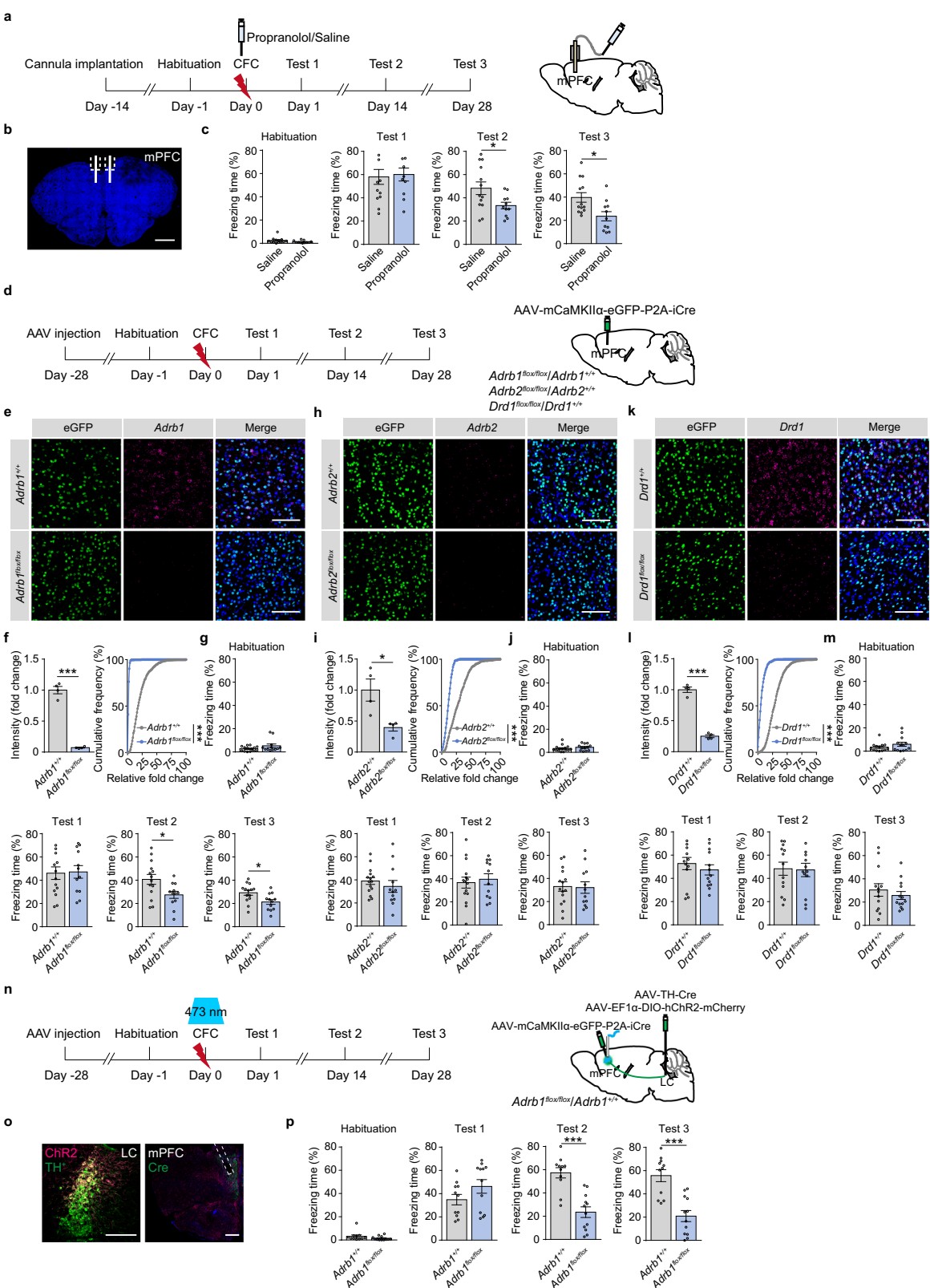

AAV$_9$-EF1α-DIO-hChR2-mCherry was generated and packaged by Obio Technology. AAV$_9$-EF1α-DIO-eNpHR3.0-EYFP, AAV$_9$-EF1α-DIO-EYFP, AAV$_9$-EF1α-DIO-mCherry, AAV$_9$-hSyn-DIO-ChrimsonR-tdTomato, AAV$_9$-mCaMKIIα-eGFP-P2A-iCre, and scAAV$_1$-hSyn-FlpO were generated and packaged by Taitool Biological. AAV$_9$-TH-Cre, AAV$_9$-TRE-tight-hM4Di-mCherry, AAV$_9$-TRE-tight-mCherry, AAV$_9$-EF1α-DIO-TVA-mCherry, AAV$_9$-EF1α-DIO-RVG, RV-ENVA-ΔG-eGFP,

AAV$_9$-hSyn-Cre, AAV$_9$-hSyn-Con-Fon-eNpHR3.0-EYFP, and AAV$_9$-hSyn-Con-Fon-EYFP were generated and packaged by BrainVTA. AAV preparation was used at a titer of 2 × 10$^{12}$ viral genomes per ml. Detailed information of viral vectors usages corresponding to each figure can be found in Supplementary Data 1-Mouse lines and viral vectors.

**Fig. 4 | LC-mPFC NEergic control of remote memory storage is dependent on β1-AR expression. a** Experimental scheme. Propranolol was bilaterally infused in the mPFC 20 minutes before CFC. **b** Representative image of the cannula above the mPFC. Scale bar: 1 mm. **c** The statistical graphs for freezing levels during memory tests. **d** Experimental scheme. *AAV-mCaMKIIα-eGFP-P2A-iCre* was injected in the mPFC of *Adrb1flox/flox*, *Adrb1+/+*, *Adrb2flox/flox*, *Adrb2+/+*, *Drd1flox/flox* and *Drd1+/+* mice. Behavioral tests were performed 1 month after virus injection. **e, f, h, i, k, l** Representative images and statistical graphs of *Adrb1*, *Adrb2*, and *Drd1* mRNA expression in the mPFC. Scale bar: 100 μm. **g, j, m** The

statistical graphs for freezing levels during memory tests. **n** Experimental scheme. *AAV-mCaMKIIα-eGFP-P2A-iCre* was injected in the mPFC of *Adrb1flox/flox* and *Adrb1+/+* mice, and *AAV-TH-Cre* mixed with *AAV-EF1α-DIO-hChR2-mCherry* was injected in the LC. Optical fibers were implanted above the mPFC. Optogenetic stimulation (473 nm, 10 mW, 20 Hz, 1 s duration in every 10 s) was delivered during CFC. **o** Representative images of ChR2-mCherry and eGFP-Cre expression and optical fiber tip. Scale bar: LC: 200 μm; mPFC: 500 μm. **p** The statistical graphs for freezing levels during memory tests. *$p < 0.05$, ***$p < 0.001$ vs indicated group.

## Stereotaxic surgery

Mice were anesthetized with 2% isoflurane and head-fixed in stereotaxic apparatus (Stoelting Instruments). Small amounts of AAVs (200 nl) were bilaterally injected (50 nl/min) in the mPFC (AP: +2.0 mm; ML: ±0.3 mm; DV: −2.2 mm), DG (AP: −1.9 mm; ML: ±1.0 mm; DV: −2.2 mm), BLA (AP: −1.5 mm; ML: ±3.3 mm; DV: −4.8 mm), NAc (AP: +1.8 mm; ML: ±1.0 mm; DV: −4.6 mm), LC (AP: −5.4 mm; ML: ±0.8 mm; DV: −3.8 mm), and vlPAG (AP: −4.7 mm; ML: ±0.5 mm; DV: −2.7 mm) with a blunt needle. The needle was under the control of a micropump (World Precision Instruments) and was kept on the target site for at least 5 min after injection for virus diffusion. Mice were allowed to recover for 3 weeks after surgery. *Adrb1flox/flox*, *Adrb2flox/flox*, and *Drd1flox/flox* mice with *AAV-mCaMKIIα-eGFP-P2A-iCre* injection underwent experiments 4 weeks after surgery in order to attain sufficient knockout efficiency.

For rabies input tracing, 200 nl of AAVs that were mixed with *AAV-EF1α-DIO-TVA-mCherry* and *AAV-EF1α-DIO-RVG* (1:1 volume mixing) were injected in the unilateral LC of *TH-Cre* mice. 100 nl *RV-ENVA-ΔG-eGFP* was injected in the ipsilateral mPFC, DG or BLA two weeks later. Mice were perfused 8 days after RV injection.

For optogenetic experiments and the recording of NE release, ceramic fiber optic cannulas (200 μm in diameter, 0.37 numerical aperture, Newdoon, Hangzhou) were implanted above the mPFC (AP: +2.0 mm; ML: ±1.1 mm; DV: −2.2 mm, 20° angle), DG (AP: −1.9 mm; ML: ±1.4 mm; DV: −2.1 mm, 10° angle), BLA (AP: −1.5 mm; ML: ±3.3 mm; DV: −4.7 mm), NAc (AP: −1.8 mm; ML: ±1.8 mm; DV: −4.6 mm, 10° angle), and the LC (AP: −5.4 mm; ML: ±1.5 mm; DV: −3.7 mm, 10° angle). Dental cement was used to cohere the cannulas to the skull.

For local propranolol or vehicle delivery, animals were implanted with a guide cannula (O.D. 0.48 mm/I.D. 0.34 mm, C.C 0.6 mm, RWD, Shenzhen) above the mPFC (AP: +2.0 mm; ML: ±0.3 mm; DV: −1.2 mm). Propranolol (10 μg/μl, 0.5 μl/side, Tocris Bioscience) or vehicle was infused into the mPFC through the injection needle (1 mm below the guide cannula, RWD, Shenzhen) slowly. Behavioral tests were performed 20 min after propranolol delivery.

## Contextual fear conditioning

Mice were placed in the conditioning chamber (MED Associates) for 5 min for adaption and the baseline freezing level was determined (Day −1, Habituation). During conditioning (Day 0), in Figs. 1–5, adult mice (P70) received three-trial footshock (0.5 mA, 1 s), and in Figs. 6, 7, juvenile (P20) and the control P70 mice received five-trial footshock in the conditioning chamber. In contextual memory tests, mice were placed in the same chamber for 2 min. The contextual memory tests were performed on Day 1 (Test 1, for recent memory), Day 14, and Day 28 (Test 2 and Test 3, for remote memory) after training[13]. Alternatively, for experiments in Fig. 2h, i, and supplementary Figs. 2, 4, a single test was performed on Day 28. For the *Fos-TRAP2* and *Arc-TRAP* mice, c-Fos immunostaining was performed 90 min after memory tests on Day 2 or Day 14. In CFC training, freezing level during conditioning was determined within a 30-second window after each footshock. In contextual memory tests, mice were placed in the same chamber for 2 min and the freezing level was determined. The freezing time and percentage during all tests were automatically analyzed by Video Freeze® software

provided by MED associates, except for those mice with optogenetic manipulation during CFC training. To avoid potential interference of the sway of patch cord connected to two optical fibers implanted in the head of the animal with behavioral analysis by the software, freezing timing for each animal during CFC training was manually determined by the researchers independently in an observer-blind fashion as suggested by previous studies[51,52].

## Open field test

Spontaneous locomotor activity was measured in an open-field (40 × 40 cm²) under 25 lux luminance for 30 min. Total distance traveled, average speed, time spent in the central arena (20 × 20 cm²), and the entries to the central arena were quantified with an automated detection system (Noldus, Wageningen, Netherlands).

## In vivo photostimulation

We connected 594 nm laser or 473 nm laser (Shanghai Dream Lasers Technology, China) to a patch cord through an Omni-directional wheel (Doric Lenses), allowing free rotation. The patch cord was then connected to the optical fibers that were implanted in the mouse brain. For photoinhibition by eNpHR3.0, a sustained 5 mW 594 nm laser was delivered 3 min before and throughout the whole conditioning period. 5 more minutes of photostimulation was applied outside the conditioning chamber. To verify the appropriate frequency of photoactivation, we used *AAV-hSyn-DIO-ChrimsonR-tdTomato* coordinate with 10 mW 594 nm laser application. 5-ms light pulses at 5, 10, 20, and 40 Hz in 1 s duration were adopted. For photoactivation by ChR2, a 10 mW 473 nm laser was delivered 20 s before and throughout the whole conditioning and 1 more minute after conditioning (5-ms light pulses at 20 Hz in 1 s duration in every 10 s).

## NE release recording and analysis

*AAV-hSyn-NE2h* was injected in the mPFC, DG, BLA, and the NAc to observe the timely norepinephrine release, and the optical fiber was implanted above the target brain area[20]. The NE fluorescence was collected with sustained 30 μW 473 nm laser and then converted to voltage signals. The MATLAB codes that we used to analysis NE fluorescence were provided in the Supplementary Software 1. We use *ΔF/F* as the parameter of statistical signal strength. For the analysis of NE release in the mPFC, DG, BLA, and the NAc pairing with the 5 footshocks, the peak value of NE2h fluorescence with a 5-s window after each footshock was collected (FS). The peak value of NE2h fluorescence with a 5-s window before the first footshock was taken as the baseline (Pre-FS1). To verify the inhibitory effect of eNpHR3.0 and the activation effect of ChrimsonR on NE release, the peak value of NE2h fluorescence with a 5-s window after each footshock or light delivery was collected[53].

## Engram labeling

To fluorescently label the neurons activated during fear conditioning, we bred the following two transgenic mice: *Fos2A-iCreER* and *ArcCreER* mice with AI14 reporter mice[54,55]. Fos-TRAP2 and Arc-TRAP mice were intraperitoneally injected with tamoxifen (125 mg/kg) 6 h after habituation and 24 h before conditioning for engram labeling. The mice were darkly housed the night before and then for 3 days following the

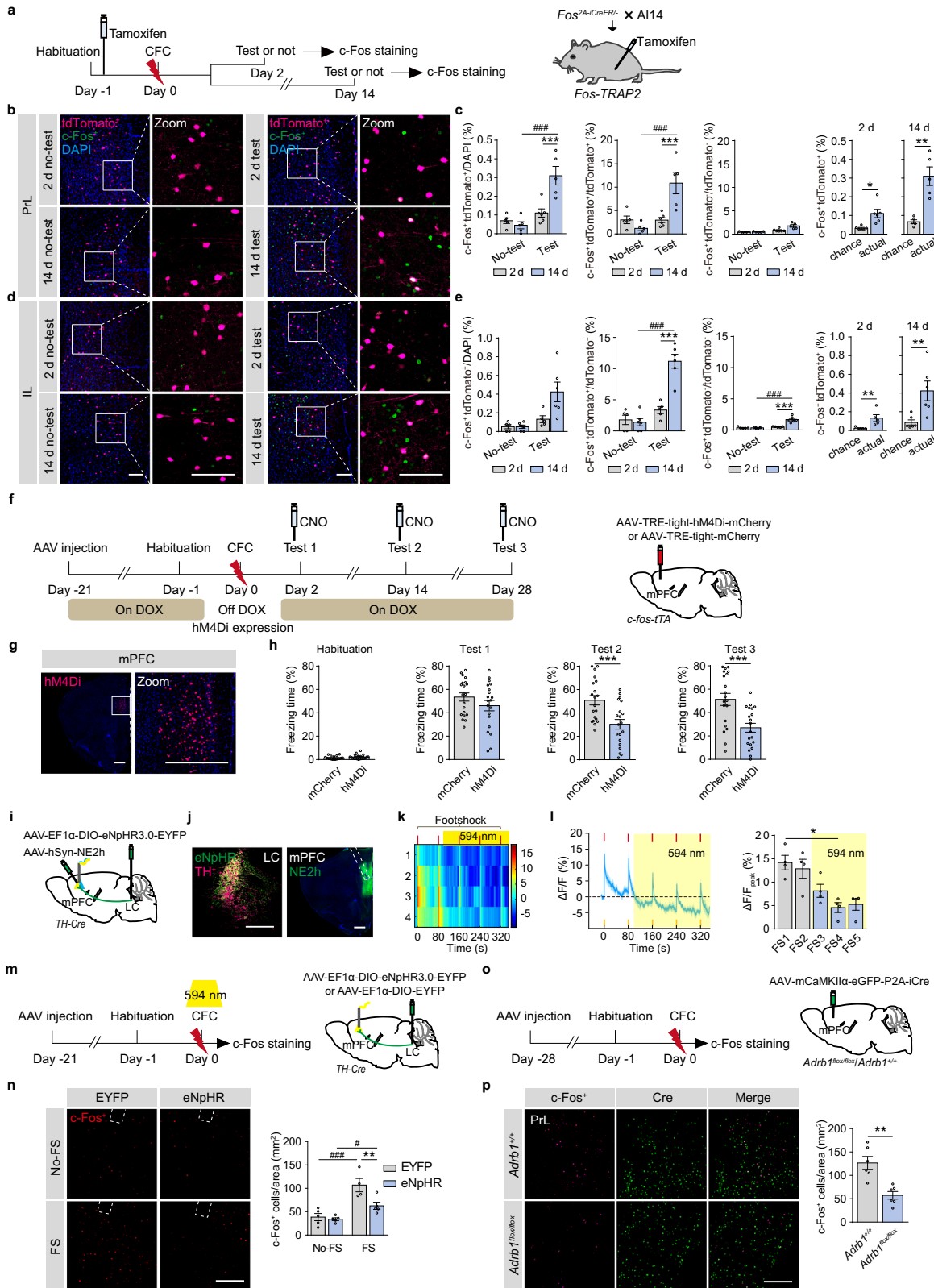

tamoxifen injection[39]. The contextual memory tests were performed 2 days or 14 days after training.

For manipulating the tagged neurons in the mPFC, which were activated by fear conditioning, *AAV-TRE-tight-hM4Di-mCherry* was injected in the mPFC of *c-fos-tTA* mice. The *c-fos-tTA* mice were fed with food containing doxycycline (40 mg/kg). 48 h before training, the food

was replaced by regular food without doxycycline in order to label the neurons that were activated by fear conditioning. Immediately after conditioning, food with doxycycline was provided. 30 min before contextual memory Test 1–3, clozapine-N-Oxide (CNO, 1 mg/kg) was intraperitoneally injected to specifically inhibit the activity of the tagged neurons in the mPFC.

**Fig. 5 | Early engram tagging in the mPFC depends on NEergic signaling.**
**a** Experimental scheme. In *Fos$^{2A-iCreER}$::AI14* mice, tamoxifen was intraperitoneally injected 24 h before CFC (125 mg/kg, i.p.). Memory tests were carried out 2 days or 14 days after CFC, and the mice were perfused 90 min after memory tests for c-Fos immunostaining. **b, d** Representative images. Scale bar: 100 μm. **c, e** The statistical graphs for double positive cell counts and percentage of c-Fos$^+$ tdTomato$^+$ and c-Fos$^+$ tdTomato$^-$ cells in the PrL (**c**) and IL (**e**). **f** Experimental scheme. *AAV-TRE-tight-hM4Di-mCherry* or *AAV-TRE-tight-mCherry* was injected in the mPFC of *c-fos-tTA* mice. A regular diet without doxycycline (off-Dox) was provided during CFC to allow *c-fos*-driven hM4Di expression. CNO (1 mg/kg, i.p.) was injected 30 min before memory tests. **g** Representative images. **h** Statistical graphs for freezing levels. **i** Experimental scheme. *AAV-EF1α-DIO-eNpHR3.0-EYFP* was injected in the LC of *TH-Cre* mice, *AAV-hSyn-NE2h* was injected in the mPFC, and an optical fiber was implanted above the mPFC. The 594 nm laser was delivered during last three footshocks. **j** Representative images. **k** The heatmap illustrates the averaged response of NE sensor (*ΔF/F%*) to each footshock. Each row of heat map represents NE2h fluorescence of a single mouse. Color scale indicates the range of *ΔF/F*. **l** Dynamic response of NE sensor to each footshock. Bar graph: The peak values of NE2h fluorescence within a 5-s window after each footshock. **m** Experimental scheme. *AAV-EF1α-DIO-eNpHR3.0-EYFP* or *AAV-EF1α-DIO-EYFP* was injected in the LC of *TH-Cre* mice, and optical fibers were implanted above the mPFC. The 594 nm laser was delivered during CFC. Mice were perfused 90 min later for c-Fos immunostaining. **n** Representative images and statistical graph for c-Fos$^+$ cell counts in the mPFC. **o** Experimental scheme. *AAV-mCaMKIIα-eGFP-P2A-iCre* was injected in the mPFC of *Adrb1$^{flox/flox}$* and *Adrb1$^{+/+}$* mice. Mice were perfused 90 min after conditioning for c-Fos immunostaining. **p** Representative images and statistical graph for c-Fos$^+$ cell counts in the mPFC. Scale bar: 200 μm, virus expression in the mPFC: 500 μm. *$p < 0.05$, #$p < 0.05$, **$p < 0.01$, ***$p < 0.001$, ###$p < 0.001$ vs indicated group.

## Immunofluorescence

Mice were anesthetized and were transcardially perfused with saline and 4% paraformaldehyde (PFA, in 0.1 M PB, pH = 7.4). The brain was immersed in 4% PFA at 4 °C for 12 h for post-fixation and then dehydrated with 20% and 30% sucrose solutions in turn. 30-μm coronal sections were sliced by cryostat microtome (Leica Instrument Co., Ltd.). Brain slices were washed with phosphate-buffered saline (0.01 M PBS) and blocked with 5% donkey serum (in 0.01 M PBST) at room temperature for 60 min. Slices were incubated with rabbit anti-c-Fos antibody (1:1000, sc-52, Santa Cruz) or mouse anti-TH antibody (1:1000, MAB318, Millipore) at 4 °C for 20 h. After being washed in PBST, the sections were incubated with secondary antibody Alexa Fluor 488 goat anti-rabbit IgG (1:1000, 111-545-144, Jackson ImmunoResearch), Cy3 goat anti-rabbit IgG (1:1000, 111-165-144, Jackson ImmunoResearch), Alexa Fluor 488 goat anti-mouse IgG (1:1000, 115-545-466, Jackson ImmunoResearch) or Cy3 goat anti-mouse IgG (1:1000, 115-165-166, Jackson ImmunoResearch) at room temperature for 90 min. For further imaging, sections were mounted with an anti-quenching mounting medium (Thermo Fisher Scientific). Images were taken by the confocal microscope (Nikon A1, NIS-AR VS.02, Japan).

## Fluorescence in situ hybridization (FISH)

To test the expression of *Adrb1*, *Adrb2* mRNA in the mPFC of the mice of different ages and to verify the knockout efficiency of *AAV-mCaMKIIα-eGFP-P2A-iCre* injected in the mPFC of *Adrb1$^{flox/flox}$*, *Adrb2$^{flox/flox}$*, and *Drd1$^{flox/flox}$* mice, we sectioned the mPFC into 10-μm coronal slices from P14, P21, P28, P42, P70 wild type mice, and *Adrb1$^{flox/flox}$*, *Adrb1$^{+/+}$*, *Adrb2$^{flox/flox}$*, *Adrb2$^{+/+}$*, *Drd1$^{flox/flox}$* and *Drd1$^{+/+}$* mice injected with *AAV-mCaMKIIα-eGFP-P2A-iCre* in the mPFC, separately. mPFC sections were incubated with probes against mouse *Adrb1*, *Adrb2*, *Drd1*, *eGFP* (*Adrb1*, [https://www.ncbi.nlm.nih.gov/nuccore/NM_007419.2/], accession No.: NM_007419.2, target region 158-1830; *Adrb2*, [https://www.ncbi.nlm.nih.gov/nuccore/NM_007420.3], accession No.: NM_007420.3, target region 55-962; *Drd1*, [https://www.ncbi.nlm.nih.gov/nuccore/NM_010076.3], accession No.: NM_010076.3, target region 444-1358; *eGFP*, accession No.: U55763.1, target region 628-1352) at 40 °C for 2 h. After being washed by washing buffer, the sections were then incubated by Amplifier 1-FL for 30 min, Amplifier 2-FL for 30 min, and Amplifier 3-FL for 15 min at 40 °C in turn. Sections were mounted with an anti-quenching mounting medium and imaged by confocal. The expression of mRNA was analyzed by custom MATLAB[56]. To detect the knockout efficiency, *Adrb1$^+$*, *Adrb2$^+$*, or *Drd1$^+$* puncta were calculated within *eGFP$^+$* neurons. To verify the *Adrb1* and *Adrb2* mRNA expression in the mPFC of different ages of mice, the puncta (*Adrb1$^+$* or *Adrb2$^+$*) in the slice were counted and were standardized by the number of DAPI. The average of 3 slices per mouse was adopted to avoid errors.

## Immunohistochemistry

We sectioned the whole LC into 30-μm serial coronal slices from P14, P21, P28, P42, and P70 mice. Slices were first washed with PBS, blocked with 5% donkey serum, and then incubated with mouse anti-TH antibody at 4 °C for 20 h. After being washed in PBS, the slices were incubated with biotin-conjugated anti-mouse IgG (1:200, 715-065-150, Jackson ImmunoResearch) at room temperature for 90 min. After washing, the sections were incubated with amplifying fluid (A:B = 1:1, freshly prepared) at room temperature for 60 min and then dyed into DAB solution for 5 min. The sections were soaked into 70%, 80%, 90%, 100% ethyl alcohol, 50% xylene (in alcohol), and 100% xylene for 5 min in turn for dehydration. Neutral resin was used to cover the sections for further imaging (Olympus DP80 Application Suite software, Cell-Sens Vl.13, Japan).

## Cell counting

To characterize the activation of neurons induced by fear conditioning or memory retrieval, mice were transcardially perfused 90 min after conditioning or contextual retrieval. To verify the number of Dbh$^+$ or TH$^+$ neurons in the LC, 30-μm serial coronal slices from *Dbh::H2B-GFP* or *TH::H2B-GFP* mice were collected. The number of c-Fos$^+$ neurons in the mPFC, DG, BLA, and GFP$^+$ neurons in the LC was calculated by Image-Pro Plus 6.0 software automatically. For c-Fos$^+$ neurons counting, the average of 3–4 slices per mouse was adopted to avoid errors. A threshold above background fluorescence was applied for cell counting. To evaluate the activation of tagged cells by memory tests, the number of tdTomato$^+$ cells, c-Fos$^+$ cells, c-Fos$^+$ tdTomato$^+$ cells, c-Fos$^+$ tdTomato$^-$ cells, and DAPI$^+$ cells in the ROI were counted. The chance level was calculated as (tdTomato$^+$/DAPI$^+$) × (c-Fos$^+$/DAPI$^+$)[11]. To calculate the volume of the LC, we measured the area of the cell body of TH$^+$ neurons in each slice and multiplied 30 μm as the volume. The counting process was performed double-blinded.

## Statistical analyses

Our experimental data were presented as mean ± s.e.m., analyzed by SPSS and MATLAB. The normality of data distribution was evaluated with Shapiro-Wilk test and homogeneity of variance was assessed by Brown-Forsythe test. All statistical tests were two-sided and adjustments were made for multiple comparisons. When normally distributed and the homogeneity of variance reached, data were analyzed with two-tailed *Student's t*-tests, one-way ANOVA or one-way ANOVA with repeated measure (RM) for multiple comparisons, and two-way ANOVA or two-way ANOVA with RM followed by post-hoc Bonferroni's test for multiple comparisons with two factors. Two-sample Kolmogorov-Smirnov test was used to compare cumulative frequency between two groups. When normality or homogeneity of variance was violated, the data were analyzed with the Mann–Whitney *U* test, two-tailed unpaired *t*-test with Welch's correction, Friedman's M test for multiple comparisons, Scheirer-Ray-Hare test, or RM ANOVA with Geisser-Greenhouse correction for multiple comparisons with two factors. The statistical significance was defined as *p* < 0.05. Detailed statistical analyses corresponding to each figure are provided in Supplementary Data 2-Statistics.

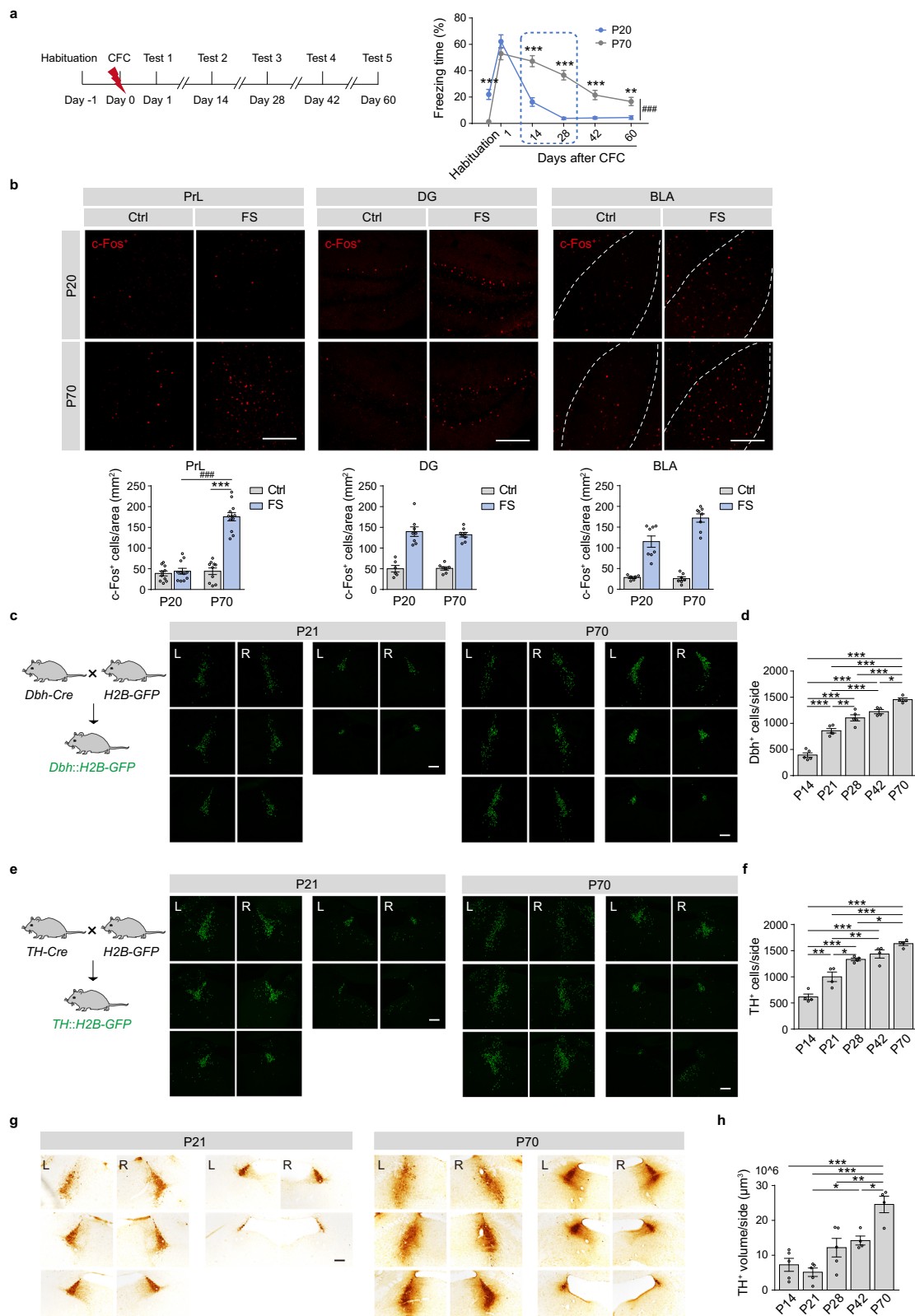

**Fig. 6 | Juvenile mice show defects in remote memory storage and immature LC/NE system. a** Experimental scheme. Fear conditioning was carried out in juvenile (P20) and adult (P70) mice, and memory tests were performed 1, 14, 28, 42, and 60 days later. **b** Representative images and statistical graphs for c-Fos⁺ cell counts in the PrL, DG, and BLA. **c, e** Representative images (every 90 μm) of H2B-GFP⁺ cells in the LC of *Dbh::H2B-GFP* mice and *TH::H2B-GFP* mice (P21 and P70). **d, f** Statistical graphs for H2B-GFP⁺ neurons in the LC of *Dbh::H2B-GFP* and *TH::H2B-GFP* mice (P14, P21, P28, P42, P70). **g** Representative images (every 90 μm) of TH immunostaining in the LC of P21 and P70 mice. **h** Statistical graph for TH volume in the LC. Scale bar: 200 μm. *$p < 0.05$, **$p < 0.01$, ***$p < 0.001$, ###$p < 0.001$ vs indicated group.

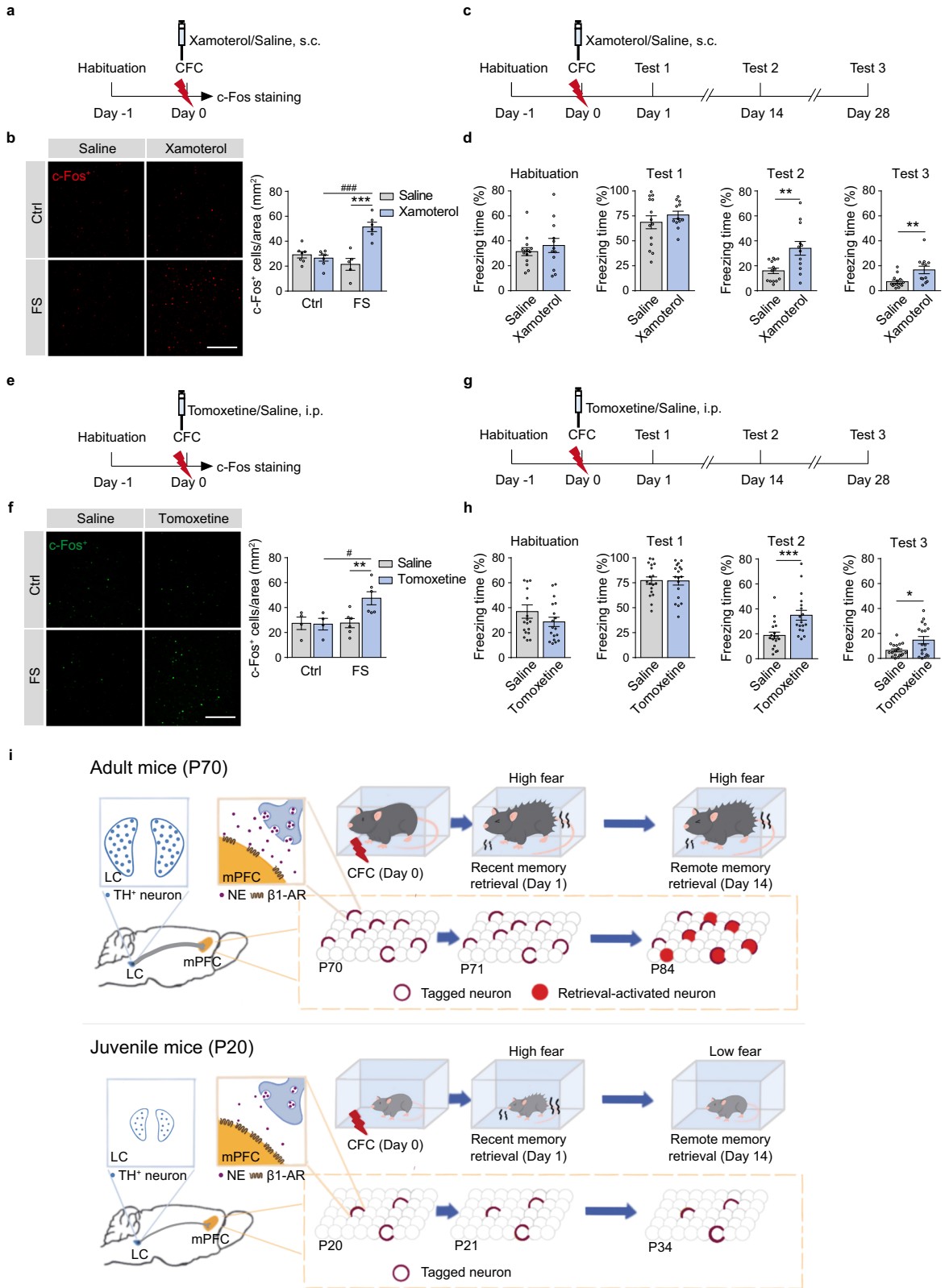

**Fig. 7 | Activation of adrenergic signaling improves remote memory storage and early engram tagging in juvenile mice. a**, **e** Experimental scheme. Xamoterol (3 mg/kg, s.c.) or tomoxetine (3 mg/kg, i.p.) was injected 30 min before CFC. The juvenile mice were perfused 90 min after CFC for c-Fos immunostaining. **b**, **f** Representative images and statistical graphs for c-Fos⁺ cell counts in the mPFC of juvenile mice. **c**, **g** Experimental scheme. Xamoterol (3 mg/kg, s.c.) or tomoxetine (3 mg/kg, i.p.) was injected 30 min before conditioning in the juvenile mice. Memory tests were performed 1, 14, and 28 days after CFC. **d**, **h** Statistical graphs for freezing levels. Scale bar: 200 μm. $*p < 0.05$, $^\#p < 0.05$, $**p < 0.01$, $***p < 0.001$, $^{\#\#\#}p < 0.001$ vs indicated group. **i** Working model illustrating that LC-mPFC NE releasing and signaling promote early cortical tagging and remote memory storage.

**Reporting summary**

Further information on research design is available in the Nature Portfolio Reporting Summary linked to this article.

## Data availability

The source data generated in this study are provided in the Supplementary Information-Source Data file. Source data are provided with this paper.

## Code availability

All custom MATLAB codes are provided in the Supplementary Software 1.

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

## Acknowledgements

We thank Dr. Yulong Li (Peking University) for providing the powerful tool for monitoring extracellular NE (AAV-hSyn-NE2h). This work was supported by grants from the Science Technology Innovation 2030 Project of China (2021ZD0203500 to F.F.W. and L.M., and 2021ZD0202104 to X.L.), the Natural Science Foundation of China (31930046 and 82021002 to L.M., 32171041 to X.L.), the CAMS Innovation Fund for Medical Sciences (2021-I2M-5-009 to L.M. and X.L.), the Shanghai Municipal Science and Technology Major Project (2018SHZDZX01 to L.M.), ZJ Lab and Shanghai Center for Brain Science and Brain-Inspired Technology, and China Postdoctoral Science Foundation (BX20180070 and 2019M661347 to X.C.F.).

## Author contributions

X.L. and L.M. designed the experiment. X.C.F., J.C.S., C.N.M., and Y.B.L. performed the behavioral experiment in adult mice and contributed to the analysis. X.C.F., J.C.S., and X.L. designed and processed engram tagging experiments and analyzed the data. X.C.F., J.C.S., C.N.M., and F.F.W. performed viral injections, brain preparation, imaging, and analysis. X.C.F., C.N.M., and J.C.S. performed behavioral tests and immunostaining in juvenile mice. L.M. and X.L. supervised the project. X.L., L.M., and X.C.F. wrote the paper.

## Competing interests

The authors declare no competing interests.
