## [Peer Review File · Nature Communications]

Noradrenergic signaling mediates cortical early tagging and storage of remote memoryREVIEWER COMMENTS

Reviewer #1 (Remarks to the Author):

The MS describes a number of studies involving combinatorial/conditional viral vectors + constructs. I counted >10 employed in the first two figures alone. The names of the genes/promoters, etc are not always commonly employed nor are they identified anywhere in the MS or supplemental materials. Further the molecular logic underlying their use is not explained nor is it properly referenced. This is especially the case for the trans-synaptic based manipulations. (on a separate note, there do not appear to be any controls for cytotoxicity that might occur over the long incubation periods for remote memory that are employed)

I am unable to complete this review without this information, lest I make a mistake in assumption of what some of the gene/promoter virus names reflect (I think it may be essential information for any of the prospective audience outside of specialists in this particular branch of functional connectomics).

Reviewer #2 (Remarks to the Author):

Fan et al. investigated the role of norepinephrine (NE) release and signaling in the mPFC, among other regions, on memory encoding and retrieval. Importantly, they studied how this is regulated at the level of early tagging of memory engram cells and how NE contributes to remote memory during development. A such, this study addresses a timely and important topic in the memory engram field using a diverse and comprehensive combination of genetic, viral and pharmacological approaches. Although I am overall enthusiastic about the manuscript, I have several suggestions for the authors to strengthen the interpretation of their the data and conclusions.

Major points:

- 1) In Fig. 1f, inhibition of LC-BLA NE projection decreased freezing levels after each footshock. Based on this, the authors conclude that the LC-BLA NE projection is required for CFC memory acquisition. However, if this is true, why do the animals show no retrieval impairment in test 1-3? In other words, how can a memory that is not acquired correctly, be subsequently retrieved without impairment?
- 2) In Figs. 1-3, animals were repeatedly tested (on day 1, day 14 and day 28). Can the authors exclude that inhibition of the LC-mPFC NE projection, or deletion of *Adr1* in the mPFC, during CFC generates a weaker memory which is more susceptible to extinction (induced by repeated testing)? Does inhibition of NE signaling in the mPFC also impair remote memory when animals are only tested on day 28?
- 3) The exp in Fig 3k-m lacks a proper control group in which memory expression is not affected.
- 4) In the engram cell tagging experiments, tamoxifen was injected 1 day prior to CFC. Why was this timepoint chosen? The long interval between tamoxifen injection and CFC increases the chance of labelling task-irrelevant neurons. For instance, can the authors exclude that neurons were labelled that were active during the habituation session? Use of the faster-acting compound 4-hydroxytamoxifen would substantially reduce the window of labelling.
- 5) Fig 4a-g: reactivation of putative engram cells (tdTomato+) should always be compared with chance level, for instance by comparing with activation of non-labelled (tdTomato-) neurons in the same region. This is particularly relevant in their study as Fos levels differ between day 2 and 14 in both the mPFC and DG.
- 6) In Fig. 4e, there appears to be very low levels of co-localization between tdTomato+ and c-Fos+ cells in the DG after recent memory retrieval (on average slightly more than 1 cell/area, and with no colocalization in 2 of the 5 animals shown). How do the authors explain this?

7) TRAP mice were used to study labelling and reactivation of engram cells and Fos-tTA mice for manipulation of engram cells. Why did the authors not also use TRAP2 mice for manipulations? The TetOff system is generally leakier, especially when combined with Cre recombinase. Did the authors verify that this system did not induce substantial non-specific labelling of neurons?

8) Fig. 5a: Why do P20 mice show significantly higher levels of freezing in the habituation session than P70 mice?

9) Fig 5b and Suppl Fig 6: Age-dependent differences in c-Fos induction seem to generalize to all brain regions examined. Does this represent reduced learning-evoked neuronal activity or perhaps a developmental difference in the threshold of c-Fos induction?

10) Given that the authors show that LC-BLA and LC-DG projections are also important for memory processing (acquisition and consolidation), how do the authors explain no deficits in acquisition and recent memory retrieval between P20 and P70 mice, despite the developmental differences in LC cell number and volume?

11) mRNA expression of *Adrb1* and *Adrb2* declines from P14-P70, which contradicts the author's conclusion that the LC/NE system is deficient in younger mice. How do the authors interpret this finding? This should be discussed.

Minor points:

12) Overall, readability of the manuscript can be improved by rigorous spelling and grammar checks.

13) Lines 49-51: please include references (Kitamura et al., 2017 Science; Matos et al., 2019 Nat Commun) for the statement that mPFC engram cells tagged during CFC training are required for remote, but not recent, memory retrieval.

14) In general, images of the mPFC do not clearly show where in the mPFC (anterior-posterior/dorsal-ventral) NE release was measured or manipulated. See for instance Fig 1d, j, l. Please provide images with a better overview of the mPFC.

15) On lines 86-89, the authors conclude that the LC-DG and LC-mPFC NE projections are required for memory consolidation and storage, respectively. However, as the manipulation was performed during encoding, the correct interpretation would be that inhibition of these NE projections during encoding impairs subsequent memory retrieval. This is further confirmed by the finding that inhibition of the LC-mPFC NE projection on post-conditioning day 1-7 or 8-14 (Suppl Fig. 2) had no effect on subsequent memory retrieval and therefore did not affect memory storage. This should be corrected throughout the manuscript.

16) Lines 109-110: This experimental approach is not clearly explained. The authors should mention that this involved anterograde transsynaptic expression of Flpo in LC neurons that receive input from the vIPAG, resulting in intersectional expression of NpHR3.0 in LC neurons that project to the mPFC.

17) The authors conclude that LC-mPFC NEergic control of remote memory storage is dependent on B1-AR signaling. As the manipulation targeted mainly excitatory neurons (based on the CaMKII promoter), they cannot exclude that B2-AR or DRD1 expression in other mPFC cell-types is involved in memory processing. This should be discussed.

18) Please specify the age of mice that were used in Fig. 1-4 in the methods section.

19) In the methods the authors state that 'Juvenile mice (P20) received five-trial footshock to acquire sufficient fear memory'. It is unclear whether a 5-shock trial was also used for P70 mice in Fig. 5a, or whether these mice received a different three-trial footshock as described in the methods?

Reviewer #3 (Remarks to the Author):

This is a monumental series of experiments showing that role of the locus coeruleus noradrenergic system in encoding, consolidation and retrieval of remote fear memory. The importance of these inputs for memory processing has been known for some time; however, these experiments provide strong corroborating evidence and add anatomical precision, using both gain and loss of function approaches. The major new finding is the demonstration that a noradrenergic 'tagging' in the PFC during acquisition is necessary for the formation of persistent long term memory. Moreover, the stronger the NE 'tag' in the PFC, the more robust the remote memory. They also showed that the tagging at acquisition was dependent upon beta adrenergic receptors. A further study showed projections from ventrolateral periaqueductal grey to all three subpopulations of LC neurons. Inhibition of this structure during training blocked the freezing behavior to the FS itself and block memory expression at all testing times.

The authors have employed a large array of state of the art techniques to address the question of differential role of forebrain LC projections. Using the currently available molecular tools they have shown that the specific roles of NE input from LC to basal lateral amygdala (BLA), Dentate Gyrus (DG) and prefrontal cortex (PFC). This being said, the manuscript requires major editing and revision in order to make it more accessible, even to a specialized public.

The first suggestion is to remove the part concerning infantile amnesia. These data and the discussion are very interesting, but distract from the central message of the paper—i.e. that LC inputs to BLA, DG and PFC are differentially involved in encoding consolidating and retrieving memories. The infantile amnesia part could be a separate paper permitting a more extensive citation of that specific literature and in depth discussion of their current results.

The experiments labelling c-fos after memory retrieval have produced some very interesting results that are presented in fig 4 and in supplementary figures. These data likewise, merit more complete presentation and further discussion. A publication of these data in a separate, short communication, would enhance their impact.

The figures could be simplified throughout. The legends should lead the reader through the figure to easily and readily understand the message.

Figure one could be separated into two. Fig 1: Footshock releases NE in the forebrain. Explain on the 'heat maps' that each row is data from a single mouse (correct?). What are the numbers on the right side of the heat map (arbitrary scale?). 'Statistical plot': are these the mean data from 5 mice? Explain ordinate. shorten the details of the statistical analyses for the bar graphs and add it to the results or supplementary information section. Fig 1 C The probe does not seem to extend into the DG; is this correct? This part of Fig 1 could also be a supplementary Figure.

The rest of Fig 1 could be a new figure: "NE release in forebrain structures for memory processes". Remove details of the statistical analyses, especially where there is obviously no difference. The stats are well described in the Methods section. Put the number of animals in the Methods and/or Results section. G could read: inhibition of NE input to BLA decreases freezing behavior during acquisition, as shown by the significant between group difference at trial3 (** $p < .001$), with no effect on memory at any time point.

I No effect of inhibition of NE input to DG on freezing during acquisition, but significant memory impairments at Day 1, 14 and 28. (add p values)

K No effect of inhibition of NE input to PFC on freezing behavior at acquisition, no effect at memory test on Day 1; significant memory deficit at day 14 and Day 28 (add p values)

M Where are the data from the Footshock combined with 20 Hz stimulation?. N there is no legend to explain this; maybe you don't need this fig. O. explain, don't just give the stats P. Q Same comments as above concerning all the stat details State Clearly no difference for memory test on day 1 but significantly stronger remote memory at Day 14 and Day 28 (indicate p values).

The remarks above pertaining to Figure 1 are relevant for all the figures in the manuscript. The authors should review each Figure carefully and first remove irrelevant information such as the statistical details, especially when there is no difference. Then make sure that all of the values on the ordinates are explained and the point of the figure is clear.

In conclusion, this is a massive work of excellent quality. The presentation should be revised and

edited to make the paper easier to read and the figures easier to decipher. The c-fos data and the infantile amnesia data each merit separate publications. To do so would lighten the current manuscript and enhance its main message.

REVIEWER COMMENTS

Reviewer #1 (Remarks to the Author):

This is a revised version of the MS titled "Noradrenergic signaling regulates cortical early tagging and storage of remote memory". The authors provide convincing evidence for the necessary role of NE from the LC, activated by vIPAG neurons during fear conditioning and activating beta1 receptors in the mPFC, for remote fear memory and engram formation in the mPFC. The issues raised by the reviewers now appear to be resolved. The use of the modified rabies virus is restricted to that of anatomical connection tracing so that toxicity even in the cells with the active virus is not an issue. Overall this is a very nice set of studies and they set the stage for the most critical issue, the examination of the mechanism responsible for the so-called "maturation" of the remote memory engram.

Point-by-point response to Reviewers:

Reviewer #1 (Remarks to the Author):

First comments (April 12th):

The MS describes a number of studies involving combinatorial/conditional viral vectors + constructs. I counted >10 employed in the first two figures alone. The names of the genes/promoters, etc are not always commonly employed nor are they identified anywhere in the MS or supplemental materials. Further the molecular logic underlying their use is not explained nor is it properly referenced. This is especially the case for the trans-synaptic based manipulations. (on a separate note, there do not appear to be any controls for cytotoxicity that might occur over the long incubation periods for remote memory that are employed)

I am unable to complete this review without this information, lest I make a mistake in assumption of what some of the gene/promoter virus names reflect (I think it may be essential information for any of the prospective audience outside of specialists in this particular branch of functional connectomics).

Response: As suggested, we have provided all the details of virus in the results and figures.

Second comments (May 24th):

This is a revised version of the MS titled "Noradrenergic signaling regulates cortical early tagging and storage of remote memory". The authors provide convincing evidence for the necessary role of NE from the LC, activated by vIPAG neurons during fear conditioning and activating beta1 receptors in the mPFC, for remote fear memory and engram formation in the mPFC. The issues raised by the reviewers now appear to be resolved. The use of the modified rabies virus is restricted to that of anatomical connection tracing so that toxicity even in the cells with the active virus is not an issue. Overall this is a very nice set of studies and they set the stage for the most critical issue, the examination of the mechanism responsible for the so-called "maturation" of the remote memory engram.

Response: We thank the help from the reviewer that have greatly improved our manuscript.

Reviewer #2 (Remarks to the Author):

Fan et al. investigated the role of norepinephrine (NE) release and signaling in the mPFC, among other regions, on memory encoding and retrieval. Importantly, they studied how this is regulated at the level of early tagging of memory engram cells and how NE contributes to remote memory during development. A such, this study addresses a timely and important topic in the memory engram field using a diverse and comprehensive combination of genetic, viral and pharmacological approaches. Although I am overall enthusiastic about the manuscript, I have several suggestions for the authors to strengthen the interpretation of their the data and conclusions.

Major points:

1) In Fig. 1f, inhibition of LC-BLA NE projection decreased freezing levels after each footshock. Based on this, the authors conclude that the LC-BLA NE projection is required for CFC memory acquisition. However, if this is true, why do the animals show no retrieval impairment in test 1-3? In other words, how can a memory that is not acquired correctly, be subsequently retrieved without impairment?

Response: We thank the reviewer for the comment. To address this issue, we have carefully examined the raw data generated by behavioral analysis software and the original videos (data in old Fig. 1 and 2). We found out that the discrepancy in Fig. 1g was originated from the behavioral analysis software we used. In the previous manuscript, all the freezing behavior is tested in the fear conditioning chamber (MED Associates) and the freezing time and percentage were measured and analyzed automatically by Video Freeze® software provided by MED associates, which is widely and successfully used for capturing and measuring freezing behavior. However, we found that in our experiments involved optical fiber implantation, the sway of two patch cords attached to the optical fiber implanted in the animal head may interfere with the capture, measurement, or analysis of freezing behavior by the software, producing erroneous data. This is in consistent with previous observations from other groups that optogenetic manipulation during habituation or test sessions interferes with the motion detection of the program, and manual scoring is suggested (Liu X, et al. Nature. 2012; Ramirez S, et al., Science. 2013). Therefore, we have re-analyzed the video recording of the experiments that involved parallel optogenetic stimulation, and have manually scored the freezing time by researchers in an observer-blind fashion. The new analysis showed that optical inhibition of LC-BLA NE projection had no significant effect on the freezing level in response to each footshock, suggesting that LC-BLA NE projection is not required for CFC acquisition (new Fig. 2c). We have replaced the freezing data involved optical manipulation with the new ones (new Figs. 2c, e, g, o, and 3e, k) in the revised manuscript. We sincerely thank the reviewer for bringing up this issue so that we can find the mistake and correct it.

*2) In Figs. 1-3, animals were repeatedly tested (on day 1, day 14 and day 28). Can the authors exclude that inhibition of the LC-mPFC NE projection, or deletion of *Adbr1* in the mPFC, during CFC generates a weaker memory which is more susceptible to extinction (induced by repeated testing)? Does inhibition of NE signaling in the mPFC also impair remote memory when animals are only tested on day 28?*

Response: We thank the reviewer for the important suggestion. As instructed, we have examined the remote memory on Day 28 after training. The mice with optogenetic inhibition of LC-mPFC

NE projection or the projection innervated by vIPAG during CFC training showed significant decrease of freezing level in memory retention test on Day 28 (Page 5, Line 95-97, Fig. 2h, i; Page 7, Line 135-137, Supplementary Fig. 2). *Adrb1* deletion in the mPFC also suppressed freezing behavior in remote memory retention test on Day 28 (Page 7, Line 153-155, Supplementary Fig. 4).

3) The exp in Fig 3k-m lacks a proper control group in which memory expression is not affected.

Response: As suggested, we have added control groups. We injected AAV-TH-Cre mixed with AAV-EF1 α -DIO-hChR2-mCherry or AAV-EF1 α -DIO-mCherry in the LC and AAV-mCaMKII α -eGFP-P2A-iCre in the mPFC of *Adrb1*^{+/+} mice (WT mice). The *Adrb1*^{+/+} mice showed intact CFC remote memory, and these *Adrb1*^{+/+} mice with optogenetic activation of LC-mPFC NE projection during CFC training showed enhanced CFC remote memory (Fig. 4n-p).

4) In the engram cell tagging experiments, tamoxifen was injected 1 day prior to CFC. Why was this timepoint chosen? The long interval between tamoxifen injection and CFC increases the chance of labelling task-irrelevant neurons. For instance, can the authors exclude that neurons were labelled that were active during the habituation session? Use of the faster-acting compound 4-hydroxytamoxifen would substantially reduce the window of labelling.

Response: We apologize for the lack of clarity of Fig. 4a (now Fig. 5a) and the description in method. *Fos-TRAP2* and *Arc-TRAP* mice were intraperitoneally injected with tamoxifen (125 mg/kg) 6 h after habituation (Page 28, Line 648-651, Fig. 5a). The neurons activated during habituation will not be labeled in this protocol. As previous study showing that the greatest TRAPed cells were detected in the visual cortex 24 h after tamoxifen injection (Guenther CJ, et al., Neuron, 2013; Denny CA, et al., Neuron, 2014), we performed CFC training 24 h after tamoxifen injection. We also chose this scheme to decrease possible effects of tamoxifen on fear conditioning. In open field task, tamoxifen injection has no significant effects on distance travelling and speed when tested 24 h later (Page 8, Line 176-178, Supplementary Fig. 5a, b). A recent study showed that activity-dependent labeling was performed by 4-OHT injection 1 h prior behavior (Roy DS, et al., Nat Commun. 2022), suggesting that 4-OHT is a better choice for activity-dependent labeling. We thank the reviewer for the suggestion, and we will try 4-OHT for *FosTRAP* or *ArcTRAP* experiments for future studies.

5) Fig 4a-g: reactivation of putative engram cells (tdTomato+) should always be compared with chance level, for instance by comparing with activation of non-labelled (tdTomato-) neurons in the same region. This is particularly relevant in their study as Fos levels differ between day 2 and 14 in both the mPFC and DG.

Response: Thanks for the important suggestion. As suggested, the ratios of c-Fos⁺ tdTomato⁺/tdTomato⁺ and c-Fos⁺ tdTomato⁻/tdTomato⁻, and percentage of double-labeling with c-Fos and tdTomato compared with the calculated chance percentages in the mPFC, DG, and BLA were presented. The data showed that ratio of c-Fos⁺ tdTomato⁺ cells in the mPFC was increased after remote memory retention test, suggesting remote memory retention test induced c-Fos expression in tdTomato⁺ cells rather than tdTomato⁻ cells (Fig. 5c, e, g, and Supplementary Figs. 5h, 6c, e, g, i).

6) In Fig. 4e, there appears to be very low levels of co-localization between *tdTomato*⁺ and *c-Fos*⁺ cells in the DG after recent memory retrieval (on average slightly more than 1 cell/area, and with no colocalization in 2 of the 5 animals shown). How do the authors explain this?

Response: Our DG data are consistent with the previous study that show low activation levels of engram cells (< 5%) after memory retrieval in *Arc-TRAP* mice (Denny CA, et al., Neuron. 2014). Other study show that activation rate of DG engram cells induced by recent CFC memory retrieval is more than 15% in *c-fos-tTA::TRE-H2B-GFP* mice. This difference may be due to different efficiency of labeling by TRAP and TetOff system.

7) TRAP mice were used to study labelling and reactivation of engram cells and *Fos-tTA* mice for manipulation of engram cells. Why did the authors not also use TRAP2 mice for manipulations? The TetOff system is generally leakier, especially when combined with Cre recombinase. Did the authors verify that this system did not induce substantial non-specific labelling of neurons?

Response: TRAP system labels cells by injection of tamoxifen that probably make the mice ill in high dose. Although we find that tamoxifen injection (125 mg/kg) has no significant effects on distance travelling and speed when tested 24 h later, we prefer TetOff system for mouse behavioral tests.

Thank you for the suggestion. To decrease the leaky effects of *AAV-TRE3g-Cre*, we have repeated the behavioral experiment in *c-Fos-tTA* mice with injection of *AAV-TRE-tight-hM4Di-mCherry* or *AAV-TRE-tight-mCherry* in the mPFC. The results showed that inhibition of hM4Di-mCherry⁺ cells in the mPFC decreased freezing levels in Test 2 and 3 on Day 14 and 28, but not in Test 1 on Day 2. We have replaced TetOff/Cre system with the TetOff/Tight system (Page 8-9, Line 188-194, Fig. 5h-j). In addition, we found that the TetTagged cell counts are less in *AAV-TRE-tight-mCherry* injection mice than those in *AAV-TRE3g-Cre* injection mice (mCherry⁺/DAPI: TRE-tight, *n* = 10, TRE3g-Cre, *n* = 8; *U* = 3, *p* = 0.0003, Mann-Whitney U test), suggesting TetOff system combined with TRE-tight is a better choice.

8) Fig. 5a: Why do P20 mice show significantly higher levels of freezing in the habituation session than P70 mice?

Response: In the study from Paul W. Frankland's lab, the juvenile and adult mice show similar baseline of freezing levels in the context without footshock (Akers KG, et al., Science, 2014,

Supplementary Materials). In others' study, the juvenile mice show greater freezing levels than the adult mice in the context before and after one-trial footshock (Li L, et al. *Neurobiol Learn Mem.* 2018). In our study, the juvenile mice showed higher freezing levels and lower locomotor activity than the adult mice in the context before fear conditioning (Page 9, Line 210-211, Supplementary Fig. 7). The different baseline of freezing levels and locomotion in the juvenile mice might be influenced by the different breeding and experimental conditions.

9) Fig 5b and Suppl Fig 6: Age-dependent differences in c-Fos induction seem to generalize to all brain regions examined. Does this represent reduced learning-evoked neuronal activity or perhaps a developmental difference in the threshold of c-Fos induction?

Response: The reviewer raises a good question. We have checked c-Fos expression in the DG and BLA of juvenile mice after fear conditioning. The data showed that conditioning significantly increased c-Fos⁺ cell counts in the DG and BLA in P20 mice (Page 9, Line 215-217, Fig. 6b).

10) Given that the authors show that LC-BLA and LC-DG projections are also important for memory processing (acquisition and consolidation), how do the authors explain no deficits in acquisition and recent memory retrieval between P20 and P70 mice, despite the developmental differences in LC cell number and volume?

Response: The hippocampus is thought to automatically and necessarily encode experiences in infants and adults (Redondo RL, et al., *Nat Rev Neurosci*, 2011; Akers KG, et al., *Science*, 2014). We found c-Fos expression was significantly increased in the DG and BLA, but not the mPFC, of juvenile mice after fear conditioning, suggesting at P20, the DG and BLA respond similar to CFC as the adults, however, the mPFC does not function as the adults. Despite the impaired remote memory of CFC, the juvenile can still form recent CFC memory. It seems memory trace can be established in some brain regions critical for recent memory formation dependent on some neurotransmitter systems or mechanisms besides the immature NE system in juvenile mice. However, immature LC-mPFC NE projection may be one of the reasons that prevent prefrontal engram generation and transfer of hippocampal memory. Once NE system sufficiently matures, it might positively contribute to memory storage through β -AR signaling in the hippocampus and amygdala, perhaps by the regulation of synaptic plasticity (Goodman AM, et al., *J Neurosci*, 2021; Seo DO, et al., *Neuron*, 2021; McCall JG, et al., *Elife*, 2017). We add this in discussion (Page 13, line 316-327).

11) mRNA expression of Adrb1 and Adrb2 declines from P14-P70, which contradicts the author's conclusion that the LC/NE system is deficient in younger mice. How do the authors interpret this finding? This should be discussed.

Response: In this study, we find that LC NE neurons and TH volume are much less in P21 mice than P70 mice, suggesting LC NE production and release may be less in juveniles than adults. Our results show that mRNA expression levels of *Adrb1* and *Adrb2* are greater in juvenile than adults, However, due to non-specificity of the commercial primary antibodies to β 1-AR and β 2-AR we have purchased, the comparisons of β -AR protein levels and membrane receptor expression levels in the mPFC between juveniles and adults are unclear. We propose that β -AR signaling in the mPFC is still at low levels, since the c-Fos expression in the mPFC was not significantly increased after fear conditioning. We have added this in discussion (Page 13, line 306-315).

Minor points:

12) Overall, readability of the manuscript can be improved by rigorous spelling and grammar checks.

Response: As suggested, we have checked spelling and grammar throughout the manuscript.

13) Lines 49-51: please include references (Kitamura et al., 2017 Science; Matos et al., 2019 Nat Commun) for the statement that mPFC engram cells tagged during CFC training are required for remote, but not recent, memory retrieval.

Response: We have included the references as suggested. Please see Page 3 Line 56.

14) In general, images of the mPFC do not clearly show where in the mPFC (anterior-posterior/dorsal-ventral) NE release was measured or manipulated. See for instance Fig 1d, j, l. Please provide images with a better overview of the mPFC.

Response: As suggested, the better images of mPFC have been provided (Figs. 1h, 2f, j, n, 3j, 4o, and 5i, l).

15) On lines 86-89, the authors conclude that the LC-DG and LC-mPFC NE projections are required for memory consolidation and storage, respectively. However, as the manipulation was performed during encoding, the correct interpretation would be that inhibition of these NE projections during encoding impairs subsequent memory retrieval. This is further confirmed by the finding that inhibition of the LC-mPFC NE projection on post-conditioning day 1-7 or 8-14 (Suppl Fig. 2) had no effect on subsequent memory retrieval and therefore did not affect memory storage. This should be corrected throughout the manuscript.

Response: As suggested, we have corrected the conclusion throughout the manuscript (Page 5, Line 99-101; Page 6, Line 113-114, 129-130; Page 7, Line 136, Page 10, Line 242-244).

16) Lines 109-110: This experimental approach is not clearly explained. The authors should mention that this involved anterograde transsynaptic expression of Flpo in LC neurons that receive input from the vIPAG, resulting in intersectional expression of NpHR3.0 in LC neurons that project to the mPFC.

Response: Thanks for the suggestion, the strategy of eNpHR3.0 expression in LC neurons that are innervated by vIPAG has been provided (Page 6-7, Line 132-135).

17) The authors conclude that LC-mPFC NEergic control of remote memory storage is dependent on B1-AR signaling. As the manipulation targeted mainly excitatory neurons (based on the CaMKII promoter), they cannot exclude that B2-AR or DRD1 expression in other mPFC cell-types is involved in memory processing. This should be discussed.

Response: We have discussed this issue as suggested. Please see Page 12, Line 283-292.

18) Please specify the age of mice that were used in Fig. 1-4 in the methods section.

Response: The information of the mouse age has been provided in the section of Contextual fear conditioning in Methods (Page 26, Line 592-593).

19) In the methods the authors state that 'Juvenile mice (P20) received five-trial footshock to acquire sufficient fear memory'. It is unclear whether a 5-shock trial was also used for P70 mice

in Fig. 5a, or whether these mice received a different three-trial footshock as described in the methods?

Response: In Fig. 6a, the adult mice (P70) also received 5-trial footshock. We have added the information in the section of Contextual fear conditioning in Methods (Page 26, Line 594-595).

Reviewer #3 (Remarks to the Author):

This is a monumental series of experiments showing that role of the locus coeruleus noradrenergic system in encoding, consolidation and retrieval of remote fear memory. The importance of these inputs for memory processing has been known for some time; however, these experiments provide strong corroborating evidence and add anatomical precision, using both gain and loss of function approaches. The major new finding is the demonstration that a noradrenergic ‘tagging’ in the PFC during acquisition is necessary for the formation of persistent long term memory. Moreover, the stronger the NE ‘tag’ in the PFC, the more robust the remote memory. They also showed that the tagging at acquisition was dependent upon beta adrenergic receptors. A further study showed projections from ventrolateral periaqueductal grey to all three subpopulations of LC neurons. Inhibition of this structure during training blocked the freezing behavior to the FS itself and block memory expression at all testing times.

The authors have employed a large array of state of the art techniques to address the question of differential role of forebrain LC projections. Using the currently available molecular tools they have shown that the specific roles of NE input from LC to basal lateral amygdala (BLA), Dentate Gyrus (DG) and prefrontal cortex (PFC). This being said, the manuscript requires major editing and revision in order to make it more accessible, even to a specialized public.

The first suggestion is to remove the part concerning infantile amnesia. These data and the discussion are very interesting, but distract from the central message of the paper—i.e. that LC inputs to BLA, DG and PFC are differentially involved in encoding consolidating and retrieving memories. The infantile amnesia part could be a separate paper permitting a more extensive citation of that specific literature and in depth discussion of their current results.

The experiments labelling c-fos after memory retrieval have produced some very interesting results that are presented in fig 4 and in supplementary figures. These data likewise, merit more complete presentation and further discussion. A publication of these data in a separate, short communication, would enhance their impact.

Response: We thank the reviewer for the suggestion and the confirmation that these results are of importance. Indeed, the roles of LC-NE in infantile amnesia and remote memory storage in adults are two topics. In addition, we find that mPFC early tagging play an important role of in remote memory storage and LC NE projections regulate engram tagging in the mPFC during memory encoding. According to the suggestion from the editor, we respectfully request to keep these data in the manuscript.

The figures could be simplified throughout.. The legends should lead the reader through the figure to easily and readily understand the message.

Figure one could be separated into two. Fig 1: Footshock releases NE in the forebrain. Explain on the ‘heat maps’ that each row is data from a single mouse (correct?). What are the numbers on the right side of the heat map (arbitrary scale?). ‘Statistical plot’: are these the mean data from 5 mice? Explain ordinate. shorten the details of the statistical analyses for the bar graphs and add it to the results or supplementary information section. Fig 1 C The probe does not seem to extend into the DG; is this correct? This part of Fig 1 could also be a supplementary Figure. The rest of Fig 1 could be a new figure: “NE release in forebrain structures for memory processes”. Remove details of the statistical analyses, especially where there is obviously no difference. The stats are

well described in the Methods section. Put the number of animals in the Methods and/or Results section.

*G could read: inhibition of NE input to BLA decreases freezing behavior during acquisition, as shown by the significant between group difference at trial3 (***) ($p < .001$), with no effect on memory at any time point.*

I No effect of inhibition of NE input to DG on freezing during acquisition, but significant memory impairments at Day 1, 14 and 28. (add p values)

K No effect of inhibition of NE input to PFC on freezing behavior at acquisition, no effect at memory test on Day 1; significant memory deficit at day 14 and Day 28 (add p values)

M Where are the data from the Footshock combined with 20 Hz stimulation?

N there is no legend to explain this; maybe you don't need this fig.

O. explain, don't just give the stats

P. Q Same comments as above concerning all the stat details State Clearly no difference for memory test on day 1 but significantly stronger remote memory at Day 14 and Day 28 (indicate p values).

Response: We thank the reviewer for the constructive suggestions. Accordingly, we have separated Fig.1 into two figures. Fig.1a-d has been put into Fig. 1 and Fig. 1e-q have been compiled in Fig. 2. In Fig. 1, each row of heat map represents the NE2h fluorescence in response to each footshock of a single mouse and the color scale on the right side indicates the range of $\Delta F/F$. The statistical plot in the left is the mean of NE2h fluorescence to each footshock from 4 to 5 mice, and the bar graph in the right is the mean of responses to 5 footshocks. We have added this information in the figure legends (Figs. 1, 3f-h, and 5k-n) and shortened the details of the statistical analyses by deleting those with no difference. All the details of statistics are included in a table. Fig. 1c (old version) showed the track of probe, so we have replaced it with a new image that showed the tip of the probe in the DG. As suggested, we have rewritten the results related to Fig.1 (Now Fig. 2, Page 5, Line 84-101) and the figure legends.

The remarks above pertaining to Figure 1 are relevant for all the figures in the manuscript. The authors should review each Figure carefully and first remove irrelevant information such as the statistical details, especially when there is no difference. Then make sure that all of the values on the ordinates are explained and the point of the figure is clear.

Response: Thanks for the suggestions. We have removed the statistics with no difference from the figure legends (Fig. 1m; Fig. 2c, e, g, o; Fig. 3e, k; Fig. 4c, g, j, m, p; Fig. 5e, j; Fig. 7d, h; Supplementary Fig. 1b, d; Supplementary Fig. 3b; Supplementary Fig. 5d, e, f, h; Supplementary Fig. 6c, e, g, i; Supplementary Fig. 13b, d) and made sure that all the ordinates have the value (Fig. 1c, f, i, l; Fig. 3g; Fig. 5m).

In conclusion, this is a massive work of excellent quality. The presentation should be revised and edited to make the paper easier to read and the figures easier to decipher. The c-fos data and the infantile amnesia data each merit separate publications. To do so would lighten the current manuscript and enhance its main message.

We thank all the help from the reviewer that have significantly improved the quality of our study.

REVIEWER COMMENTS

Reviewer #2 (Remarks to the Author):

Fan et al. did an excellent job in addressing the main concerns and suggestions of the reviewers and the manuscript has improved substantially. However, I still have some minor points and questions that relate to my previous comments (and their numbering).

3) For Fig 4n-p, the authors added *Adrb*^{+/+} mice to include control groups that have intact memory, as requested. However, I have a minor issue with the statistical presentation of the data. The authors describe that the mouse x virus interaction is not significant in the ANOVA, but they do show post-hoc comparisons. Without describing potential significant main effects of mouse and/or virus, post-hoc comparisons are not valid. In this experiment, a main effect of factor 'mouse' is sufficient to demonstrate a lack of rescue after deletion of *Ardb1* and this will likely be significant based on the differences between groups. Also, post-hoc comparison reveals a difference between the *Ardb1*^{+/+} mice groups, but this comparison is only allowed when ANOVA shows a main effect of virus. Hence, this information is required to conclude that memory was enhanced in control mice with ChR2. Please correct this by describing the significant main effects of the ANOVA (if they are present) and removing non-significant factors/interactions (as suggested by reviewer 3) and potential irrelevant post-hoc comparisons.

The same argument applies to other datasets, such as in Fig. 5.

6) Given the relatively low level of reactivation in the DG, with 2 mice showing no reactivation at all, it is questionable whether the tagged population fully represents the engram population in this region. The authors acknowledge that this may be due to labeling efficiency. This should be briefly discussed, or the authors can also decide to remove the DG dataset (Fig. 5d-e) as their main focus is on the mPFC and this DG data is not relevant for their main conclusions.

7) I appreciate that the authors changed their intervention method to reduce leakiness in the exp of Fig 5h-j by replacing the TRE-Cre vector with a non-Cre TRE-vector in the *c-fos*-tTA mice. However, with the non-Cre system, expression of the DREADD or reporter is transient and previous work showed that expression is fully diminished one month after cells were labelled with a similar TetOff system (Liu et al., 2012 Nature). Can the authors provide evidence that DREADD expression is still present at day 28 after labelling?

8) The authors now mention the difference in baseline freezing between juvenile and adult mice in the text, and provide a potential explanation in the rebuttal, but not in the manuscript. Moreover, I do not understand why they argue that different breeding and experimental conditions may underlie the baseline differences as one would expect that these particular factors should not differ between juvenile and adult mice. I acknowledge that this baseline difference does not affect interpretation of their findings and that differences in anxiety levels between juvenile and adult mice may explain their observation. However, it is relevant to add a brief explanation to the manuscript.

11) I can follow the argument of the authors that the difference in mRNA levels of *Adrb1/2* may not translate to the same difference in protein levels and membrane expression, but please explain this more clearly in the discussion.

12) There are still several sentences that are grammatically not correct and need to be rephrased. For instance, see:

Line 84-85, To determine how LC NEergic afferents recruit in CFC memory storage...

line 247-249, "...mPFC memory engram generated dependent on NE release and β 1-AR signaling during initial learning mediated remote memory storage."

Reviewer #3 (Remarks to the Author):

The revision is somewhat improved from the original manuscript, but still suffers from being very difficult to follow, except for a very specialized audience. An initial reviewer (reviewer 1 (April 12) requested that the molecular terminology --viral vectors, constructs, genes, promoters, etc--be defined and the rationale for their use be explained and referenced. This has been attempted for the new version, but only to a limited extent. A complete glossary of all the molecular terms, viral vectors, constructs, mouse lines etc should be included along with the appropriate references and rationale for their use, in the form of a Table. This would greatly facilitate accessibility and comprehension for a wider audience of interested readers.

The Figure legends remain over loaded with unnecessary details, without a clear statement of the message of the figure in its legend. The statistical methods are well-described within Methods section, so no need to include all the test details in Fig legends. To make the Fig legends more readable, state the main point of the figure and limit stat information to t, U or F values, dfs and p values.

This reviewer is still convinced that the work would have more impact if the infantile amnesia work were presented as a follow up complementary paper. This would allow for a more extended discussion focused on the main results.

The paper still requires editing for grammar. Even the addition of a few commas would facilitate readability (e.g. line 247 and many others).

In conclusion, as was noted in my initial review, the body of work is impressive, using state of the art molecular tools to corroborate a large body of earlier work showing a role for the noradrenergic system in early and remote memory processes. More effort should be made in the presentation, to increase accessibility to the wide neuroscience audience this work merits.

Point-by-point response to Reviewer #2 and #3:

Reviewer #2 (Remarks to the Author):

Fan et al. did an excellent job in addressing the main concerns and suggestions of the reviewers and the manuscript has improved substantially. However, I still have some minor points and questions that relate to my previous comments (and their numbering).

3) For Fig 4n-p, the authors added Adrb^{+/+} mice to include control groups that have intact memory, as requested. However, I have a minor issue with the statistical presentation of the data. The authors describe that the mouse x virus interaction is not significant in the ANOVA, but they do show post-hoc comparisons. Without describing potential significant main effects of mouse and/or virus, post-hoc comparisons are not valid. In this experiment, a main effect of factor 'mouse' is sufficient to demonstrate a lack of rescue after deletion of Ardb1 and this will likely be significant based on the differences between groups. Also, post-hoc comparison reveals a difference between the Ardb1^{+/+} mice groups, but this comparison is only allowed when ANOVA shows a main effect of virus. Hence, this information is required to conclude that memory was enhanced in control mice with Chr2. Please correct this by describing the significant main effects of the ANOVA (if they are present) and removing non-significant factors/interactions (as suggested by reviewer 3) and potential irrelevant post-hoc comparisons. The same argument applies to other datasets, such as in Fig. 5.

Response: We would like to start by thanking Reviewer #2 for the confirmation and encouragement and we appreciate the time spent for reviewing this manuscript.

As suggested, we have corrected the description, removing non-significant interactions, and potential irrelevant post-hoc comparisons in Fig. 4p and Fig. 5c and 5e (Page 8, Line 164-166, and 177-187, in Fig. 4p, Fig.5b-e, and Supplementary Figs. 5 and 6).

6) Given the relatively low level of reactivation in the DG, with 2 mice showing no reactivation at all, it is questionable whether the tagged population fully represents the engram population in this region. The authors acknowledge that this may be due to labeling efficiency. This should be briefly discussed, or the authors can also decide to remove the DG dataset (Fig. 5d-e) as their main focus is on the mPFC and this DG data is not relevant for their main conclusions.

Response: Thank you for the suggestion. We have discussed about the labeling efficiency in the DG of TRAP system and removed the DG and BLA datasets to the supplementary figures (Page 12, Line 293-301, Supplementary Fig. 5c-i).

7) I appreciate that the authors changed their intervention method to reduce leakiness in the exp of Fig 5h-j by replacing the TRE-Cre vector with a non-Cre TRE-vector in the c-fos-tTA mice. However, with the non-Cre system, expression of the DREADD or reporter is transient and previous work showed that expression is fully diminished one month after cells were labelled with a similar TetOff system (Liu et al., 2012 Nature).

Can the authors provide evidence that DREADD expression is still present at day 28 after labelling?

Response: Thank you for the suggestion. We checked the expression of hM4Di-mCherry in the mPFC after remote memory retention test 3 (Day 28), as shown in Fig. 5g.

8) The authors now mention the difference in baseline freezing between juvenile and adult mice in the text, and provide a potential explanation in the rebuttal, but not in the manuscript. Moreover, I do not understand why they argue that different breeding and experimental conditions may underlie the baseline differences as one would expect that these particular factors should not differ between juvenile and adult mice. I acknowledge that this baseline difference does not affect interpretation of their findings and that differences in anxiety levels between juvenile and adult mice may explain their observation. However, it is relevant to add a brief explanation to the manuscript.

Response: As suggested, we have added a brief explanation in the Discussion (Page 13, Line 325-328).

11) I can follow the argument of the authors that the difference in mRNA levels of Adrb1/2 may not translate to the same difference in protein levels and membrane expression, but please explain this more clearly in the discussion.

Response: We have explained the difference in mRNA levels of *Adrb1/2* in the Discussion as suggested (Page 13, Line 319-324).

12) There are still several sentences that are grammatically not correct and need to be rephrased. For instance, see:

Line 84-85, To determine how LC NEergic afferents recruit in CFC memory storage... line 247-249, "...mPFC memory engram generated dependent on NE release and β 1-AR signaling during initial learning mediated remote memory storage."

Response: Thank you for the suggestions. We have rephrased these sentences (Page 5, Line 83-84; Page 11, Line 246-249) and edited the manuscript for grammar (The changes are highlighted with colored fronts).

Reviewer #3 (Remarks to the Author):

The revision is somewhat improved from the original manuscript, but still suffers from being very difficult to follow, except for a very specialized audience. An initial reviewer (reviewer 1 (April 12) requested that the molecular terminology --viral vectors, constructs, genes, promoters, etc-- be defined and the rationale for their use be explained and referenced. This has been attempted for the new version, but only to a limited extent. A complete glossary of all the molecular terms, viral vectors, constructs, mouse lines etc should be included along with the appropriate references and rationale for their use, in the form of a Table. This would greatly facilitate accessibility and comprehension for a wider audience of interested readers.

Response: We appreciate the encouragement from the Reviewer #3 and thank for constructive advices.

Thank you for the suggestion. A glossary of viral vectors and mouse lines used has been listed in a table uploaded as a supplementary file.

The Figure legends remain over loaded with unnecessary details, without a clear statement of the message of the figure in its legend. The statistical methods are well-described within Methods section, so no need to include all the test details in Fig legends. To make the Fig legends more readable, state the main point of the figure and limit stat information to t, U or F values, dfs and p values.

Response: Thank you for the suggestion. We have shortened the figure legends and removed the statistical information with no significant interaction.

This reviewer is still convinced that the work would have more impact if the infantile amnesia work were presented as a follow up complementary paper. This would allow for a more extended discussion focused on the main results.

Response: We thank the reviewer for the suggestion and the confirmation that the data from the juvenile mice are meaningful. We propose that the infantile amnesia work will strengthen our conclusion that mPFC early tagging play an important role of in remote memory storage and LC NE projections regulate engram tagging in the mPFC during memory encoding. We respectfully request to keep these data in the manuscript.

The paper still requires editing for grammar. Even the addition of a few commas would facilitate readability (e.g. line 247 and many others).

Response: Thank you for the suggestion. We have edited the manuscript for grammar (The changes are highlighted with colored fronts).

In conclusion, as was noted in my initial review, the body of work is impressive, using state of the art molecular tools to corroborate a large body of earlier work showing a role for the noradrenergic system in early and remote memory processes. More effort should be made in the presentation, to increase accessibility to the wide neuroscience audience this work merits.

Thank you for the confirmation and advices.